# HoloScene: Simulation-Ready Interactive 3D Worlds from a Single Video

**Hongchi Xia**[1]  **Chih-Hao Lin**[1]  **Hao-Yu Hsu**[1]
**Quentin Leboutet**[2]  **Katelyn Gao**[2]  **Michael Paulitsch**[2]
**Benjamin Ummenhofer**[2]  **Shenlong Wang**[1]

[1]University of Illinois Urbana-Champaign    [2]Intel

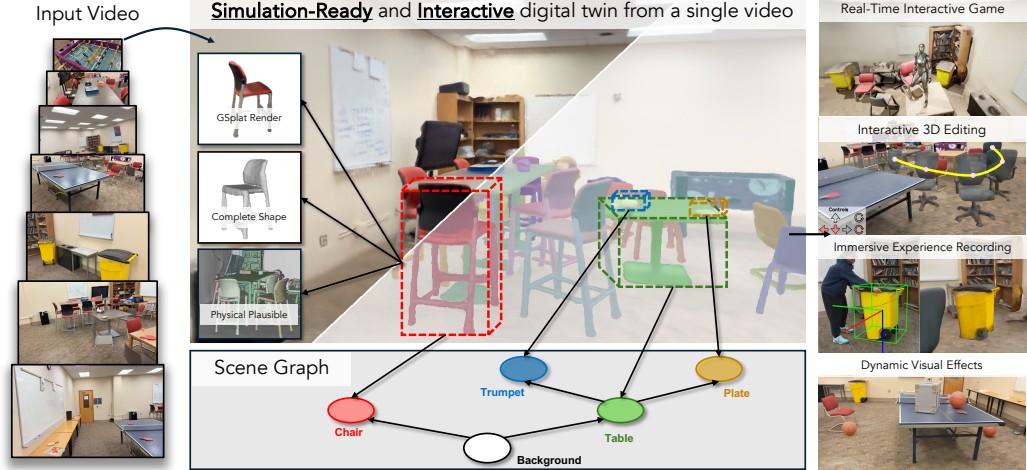

Figure 1: **Overview of HoloScene:** From a single input video—along with visual cues such as segmentation and monocular depth—HoloScene reconstructs a simulation-ready, interactive 3D digital twin represented as a scene graph with complete geometry, physically plausible dynamics, and realistic rendering. The resulting model enables a variety of downstream applications, including real-time interactive gaming, 3D editing, immersive experience capture, and dynamic visual effects.

## Abstract

Digitizing the physical world into accurate simulation-ready virtual environments offers significant opportunities in a variety of fields such as augmented and virtual reality, gaming, and robotics. However, current 3D reconstruction and scene-understanding methods commonly fall short in one or more critical aspects, such as geometry completeness, object interactivity, physical plausibility, photorealistic rendering, or realistic physical properties for reliable dynamic simulation. To address these limitations, we introduce HoloScene, a novel interactive 3D reconstruction framework that simultaneously achieves these requirements. HoloScene leverages a comprehensive interactive scene-graph representation, encoding object geometry, appearance, and physical properties alongside hierarchical and inter-object relationships. Reconstruction is formulated as an energy-based optimization problem, integrating observational data, physical constraints, and generative priors into a unified, coherent objective. Optimization is efficiently performed via a hybrid approach combining sampling-based exploration with gradient-based refinement. The resulting digital twins exhibit complete and precise geometry, physical stability, and realistic rendering from novel viewpoints. Evaluations conducted on multiple benchmark datasets demonstrate superior performance, while practical use-cases in interactive gaming and real-time digital-twin manipulation illustrate HoloScene's broad applicability and effectiveness. Project page: here.

39th Conference on Neural Information Processing Systems (NeurIPS 2025).

# 1 Introduction

Imagine wanting, decades later, to revisit the home you live in and love today—how would you capture its memory? Photographs and videos record authentic details but lack immersion; 3D Gaussian splats or photogrammetry can be immersive, yet static chairs and tables feel lifeless. Ideally, we would digitize our environment into a fully interactive digital twin: complete, composable, photorealistic, and manipulable just like the real world. Our work takes a step toward this goal by enabling users to create in silico twins of their surroundings from a single video.

Digitizing the physical world into a simulation-ready virtual environment offers immense opportunities in augmented and virtual reality, gaming, and robotics. However, despite advances in 3D modeling and scene understanding, key challenges remain: capturing complete geometry and appearance in occluded regions, inferring inter-object relationships, and ensuring physical plausibility and interactivity. Existing Real2Sim methods produce incomplete geometry [88, 45, 71] or unstable physics [77, 75]; existing amodal reconstruction focuses on single-image setting [86, 11], individual objects [76, 15, 39], neglects physical plausibility [49] or relies on asset retrieval [10] —sacrificing fidelity and practicality; and prior physically plausible reconstruction [48, 15] is limited to simple object–scene interactions or requires full observations.

To address these gaps, we introduce **HoloScene**, an interactive 3D reconstruction framework that unifies geometry completeness, object completeness, physical plausibility, realistic rendering, and physical interaction. HoloScene represents a scene as an interactive scene graph encoding object geometry, appearance, and physical properties in a hierarchical structure. We cast scene-graph recovery from video as a structured energy-based optimization, integrating observational data, physical constraints, and generative priors into a single objective. To solve this challenging problem, we propose a novel divide-and-conquer strategy combining sampling-based tree-structured search with gradient-based refinement. The resulting scene models exhibit complete, accurate geometry; stable physical interactions; and realistic rendering from novel viewpoints.

Experiments on three challenging benchmarks demonstrate superior geometry accuracy and physical plausibility, with rendering performance comparable to state-of-the-art amodal and physics-aware reconstruction methods. We further showcase HoloScene's versatility through practical applications in interactive gaming, realistic video effects, and real-time digital-twin manipulation.

# 2 Related Works

**Interactive 3D Scene Model** Recent advances in 3D scene modeling [52, 93, 17, 5, 68] reconstruct 3D scene from input images or videos, representing the scene as neural fields [46, 65, 19, 62, 61, 25, 3, 4], signed distance functions (SDF) [71, 87, 51, 53, 88, 91, 29], and 3D Gaussians [23, 20, 92, 90]. While producing realistic renderings from novel views, these works cannot provide 3D assets that allow user

| Method | Visual Input | Real-time Rendering | Amodal 3D Recon | Twin Fidelity | Physics Capacity | Physics Optimization |
|---|---|---|---|---|---|---|
| ACDC [10] | image | ✓ | ✓ | ✗ | ✗ | ✗ |
| Gen3DSR [11] | image | ✓ | ✓ | ✓✗ | ✗ | ✗ |
| PhysComp [15] | image | ✗ | ✓ | ✓✗ | Single Object | Differentiable |
| CAST [86] | image | ✓ | ✓ | ✓✗ | Scene | Differentiable |
| NeRF [45] | video | ✗ | ✗ | ✗ | ✗ | ✗ |
| BakedSDF [88] | video | ✗ | ✗ | ✓ | ✗ | ✗ |
| ObjectSDF++ [75] | video | ✗ | ✗ | ✓ | ✗ | ✗ |
| Video2Game [77] | video | ✓ | ✗ | ✓ | Single Object | ✗ |
| DRAWER [78] | video | ✓ | ✗ | ✓ | Single Object | ✗ |
| PhyRecon [48] | video | ✗ | ✓ | ✓ | Objects-Ground | Differentiable |
| DP-Recon [49] | video | ✓ | ✓ | ✓ | ✗ | ✗ |
| **HoloScene (Ours)** | video | ✓ | ✓ | ✓ | Scene | Diff & Sampling |

Table 1: **Comparison of Interactive 3D Scene Models.**

interactions (e.g, move the chairs to different poses). Reconstructing realistic and interactive environments from real images and videos remains challenging due to limited observation, occlusion, and physical reasoning. Some previous works reconstruct 3D objects from sparse viewpoints [39, 35, 79, 7, 41], and some estimate physical properties from visual observation [94, 15]. Nevertheless, these works focus on object-level tasks and cannot handle large and complex indoor scenes. PhyRecon [48] optimizes stable 3D scenes with differentiable physical engines, but does not model inter-object interactions and realistic appearance. DP-Recon [49], Video2Game [77], Drawer [78] leverage generative prior [1, 55] and foundation models [59, 60] to reconstruct decomposed 3D scenes. However, these works can only produce limited components or lack physical stability. Text-to-3D generation [83, 33, 72] creates diverse layouts from text prompts, but fails to reconstruct digital twins that are faithful to visual inputs. In this work, we propose to reason

object interaction with the scene graph, and utilize generative priors and a novel sampling strategy to reconstruct the geometry and appearance of every component, constructing realistic, physically plausible, and interactable 3D environments.

**Data-driven Simulation**  Simulation plays a pivotal role across robotics, self-driving, and content creation, but building high-fidelity virtual scenes remains costly, and the sim-to-real gap poses great challenges. To address this, data-driven simulation [2, 8, 40, 44, 63, 84] has emerged, enabling the modeling of physical dynamics [38, 28, 80, 21, 13, 22, 95], lighting conditions [32, 58, 31], and action-conditioned outcomes [21, 36, 6, 77, 22, 37, 78, 42], directly from real-world data. These methods have also been applied in robot learning [8, 44, 57, 82, 85, 84], LiDAR simulation [34, 44, 81, 84, 99, 98], and interactive media [18, 77]. In robotics, related *real-to-sim* approaches [9, 10, 67, 70, 27, 43] reconstruct interactable environments from the real world for reproducible embodiment. However, they still lack physical realism. Recent works [48, 6, 97] leverage differentiable physics or priors in reconstruction, but they neglect complex inter-object relationships. The closest work to ours is CAST [86], which also considers physical plausibility in scene reconstruction. However, our proposed HoloScene differs fundamentally in three key aspects: (1) Problem formulation: CAST takes a single-image generative approach where fidelity to input appearance and geometry is not enforced, while HoloScene performs multi-view joint optimization from videos, explicitly ensuring faithful reconstruction of observed scenes through inverse neural rendering. (2) Physical stability: CAST's differentiable optimization only avoids penetrations but cannot guarantee long-term stability. HoloScene employs simulator-in-the-loop optimization with Isaac Sim [47] to ensure objects remain physically stable over time. (3) Inference strategy: CAST follows a multi-step pipeline with greedy sequential optimization, whereas HoloScene adopts a unified energy-based formulation (Eq. 2) that jointly optimizes observation fidelity, physical plausibility, and object completion through generative sampling and scoring. We compare HoloScene with prior works in Tab. 1.

## 3 Method

Given the observations $\mathcal{O} = \{\mathcal{O}_t\}_{t=0}^T$, which include the input video sequence $\{\mathbf{I}_t\}$, camera poses $\{\boldsymbol{\xi}_t\}$ (inferred or ground truth), and instance masks $\{\mathbf{M}_t\}$ (inferred or ground truth), our goal is to reconstruct a realistic, complete, and physically plausible digital twin of the input scene, yielding interactive, sim-ready assets compatible with simulators and game engines, and which can be used to generate novel visual content. To this end, we represent the scene as an interactive 3D scene graph representation that encodes object geometry, appearance, physical properties, and hierarchical inter-object relationships (Sec. 3.1). We combine observational evidence, generative priors for shape completion, and physical simulation for stability to formulate scene-graph recovery as an energy minimization problem (Sec. 3.2). Finally, we propose an inference method that integrates sampling-based tree search with differentiable optimization (Sec. 3.3). Fig.2 summarizes our approach.

### 3.1 Scene Representation

We represent the scene as an interactive 3D scene graph $\mathcal{G} = (\mathcal{V}, \mathcal{E})$. Each node $\mathbf{v}_i \in \mathcal{V} = \{\mathbf{v}_i\}_{i=0}^N$ represents either the background scene or one of the $N$ objects present. A node $\mathbf{v}_i = (\mathbf{g}_i, \mathbf{f}_i, \mathbf{p}_i, \mathbf{T}_i)$ is comprised of geometry $\mathbf{g}_i$, appearance $\mathbf{f}_i$, physical properties $\mathbf{p}_i$, and dynamic states $\mathbf{T}_i$. Each edge $\mathbf{e}_{i,j} = (\mathbf{v}_i, \mathbf{v}_j) \in \mathcal{E}$ encodes an object–object relationship in $\mathcal{G}$.

**Geometry:** We represent the geometry of each node $\mathbf{v}_i$ in the scene with an instance-level neural SDF $g_i(\mathbf{x}; \theta)\colon \mathbb{R}^3 \to \mathbb{R}$, where $\mathbf{x} \in \mathbb{R}^3$ is any point in space and $\theta$ are learnable parameters. Additionally, to facilitate physical simulation and efficient rendering, we maintain a mesh representation $\mathcal{M}_i = \texttt{MarchingCube}(\mathbf{g_i})$ for each object, extracted from its SDF using the marching cubes algorithm.

**Appearance:** For each object $\mathbf{v}_i$, we encode appearance $\mathbf{f}_i = (\mathbf{c}_i, \alpha_i, \boldsymbol{\mu}_i, \Sigma_i)$ as Gaussian splats, enabling real-time, high-quality rendering. $\mathbf{c}_i, \alpha_i, \boldsymbol{\mu}_i, \Sigma_i$ are color, opacity, mean, and covariance of Gaussians, respectively. Gaussians capture finer detail than colored meshes but hinder consistency between appearance, geometry, and simulation; following recent work [78], we adopt a Gaussians-on-Mesh (GoM) approach and attach each splat to its mesh to ensure alignment and enable physical interactions. Given camera intrinsics $\mathbf{K}$ and extrinsics $\boldsymbol{\xi}$, we denote the splat-rendered RGB images, masks, depth and normal maps as $\mathbf{I}, \mathbf{M}, \mathbf{D}, \mathbf{N} = \texttt{SplatRender}(\mathcal{G}; \mathbf{K}, \boldsymbol{\xi})$.

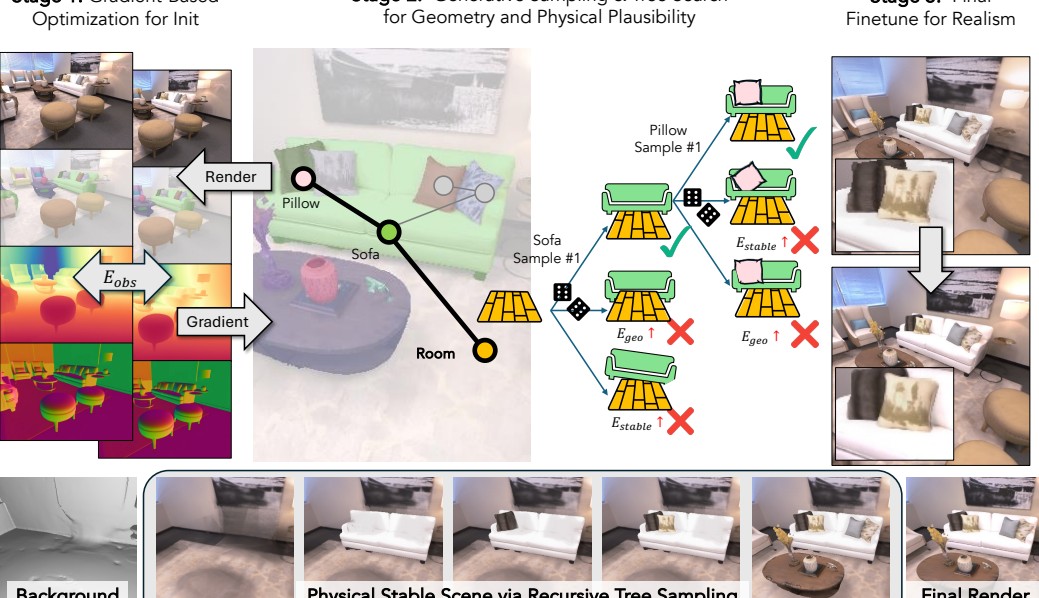

**Stage 1:** Gradient-Based Optimization for Init

**Stage 2:** Generative Sampling & Tree Search for Geometry and Physical Plausibility

**Stage 3:** Final Finetune for Realism

Render
$E_{obs}$
Gradient
Pillow
Sofa
Room

Pillow Sample #1
Sofa Sample #1
$E_{stable}$ ↑
$E_{geo}$ ↑
$E_{geo}$ ↑
$E_{stable}$ ↑

Background
Physical Stable Scene via Recursive Tree Sampling
Final Render

Figure 2: **Overview of HoloScene Optimization Stages:** Given multiple posed images as well as some visual cues (instance masks, monocular geometry priors), we first employ a gradient-based optimization as the initialization. Then we adopt a generative sampling and tree search strategy along the topology of the scene graph to obtain the complete geometry with physical plausibility. Finally, the final fine-tuning over the scene further enhances the realism of the reconstructed scene.

**Physics:** Each object in our scene graph is modeled as a rigid body. Its physical parameters $\mathbf{p}_i = (m_i, \kappa_i, \zeta_i, r_i)$ comprise mass $m_i$, friction $\kappa_i$ (resistance to sliding against other surfaces), damping $\zeta_i$ (energy dissipation during motion), and restitution $r_i$ (elasticity upon impact). These parameters are used in downstream physical simulations to model the object's response to external forces and its interactions with other objects and the background scene.

**Object States:** All object intrinsic attributes above are defined in object-centric coordinates and remain invariant under motion. To handle dynamic changes, we encode each object's rigid body state by a rigid transform $\mathbf{T}_i$ from its object-centric frame to the world frame. During static reconstruction, $\mathbf{T}_i$ is fixed; in dynamic simulation, it may vary over time. Let $\mathcal{T} = \{\mathbf{T}_i\}_{i=0}^N$ denote the set of all object states, $\mathcal{G}_\mathcal{T}$ the scene graph under those states, and $\mathcal{G}$ the scene graph under the static state during reconstruction.

**Object Relationships:** Each edge $\mathbf{e}_{i,j}$ links two nodes with one of three relationships: 1) `support`, where $\mathbf{v}_i$ rests in stable equilibrium on its unique parent $\mathbf{v}_{\mathrm{pa}(i)}$ under gravity (each object has exactly one such parent, so support edges form a tree in the static scene graph); 2) `beside`, where siblings $(\mathrm{pa}(j) = \mathrm{pa}(i))$ have touching surfaces, causing occlusions without hierarchy or instability; and 3) `collide`, where contacts with nonzero momentum yield dynamic effects—ignored during static reconstruction but employed in simulation. Note that the object relationship might change depending on its dynamic status during simulation.

**Interaction & Simulation:** Our 3D scene graph's distinguishing feature is its support for physical interactions. Formally, at time $t$, given the dynamic scene graph $\mathcal{G}_{\mathcal{T}^t}$ with current object states $\mathcal{T}^t$ as well as an input action $\mathbf{a}^t$, the next states are computed as

$$\mathcal{T}^{t+1} = \mathtt{Sim}(\mathcal{T}^t, \mathbf{a}^t; \mathcal{G}_{\mathcal{T}^t}), \tag{1}$$

where `Sim` is a rigid-body physical simulator using the mesh $\{\mathcal{M}_i\}$ as collision geometry. Here, $\mathbf{a}^t$ can represent external inputs— forces, torques, or control actions—applied to the objects at time $t$.

### 3.2 Problem Formulation

Our framework takes input observations $\mathcal{O}$ of a static scene and recovers the scene graph $\mathcal{G} = (\mathcal{V}, \mathcal{E})$. The resulting scene graph must (i) explain the observations well; (ii) be geometrically complete and plausible; and (iii) reflect the scene's static, physically stable nature. To this end, we cast the problem as a structured energy-minimization problem:

$$\min_{\mathcal{G}} \underbrace{E_{\text{rgb}}(\mathcal{I}, \mathcal{G}) + E_{\text{mask}}(\mathcal{M}, \mathcal{G}) + E_{\text{mono}}(\mathcal{D}, \mathcal{G})}_{\text{observation terms}} + \underbrace{E_{\text{comp}}(\mathcal{G}) + E_{\text{geo}}(\mathcal{G}) + E_{\text{physics}}(\mathcal{G})}_{\text{regularization terms}}. \quad (2)$$

For simplicity, we omit the hyperparameter linear weights for each term. Next, we discuss each energy term.

**Observation Terms:** The observation terms quantify the discrepancy between the reconstructed 3D scene and the input observations. Let $\hat{\mathbf{I}}_t, \hat{\mathbf{M}}_t, \hat{\mathbf{D}}_t, \hat{\mathbf{N}}_t = \texttt{SplatRender}(\mathcal{G}; \mathbf{K}, \boldsymbol{\xi}_t)$ denote the rendered RGB image, instance mask, depth map, and normal map at camera pose $\boldsymbol{\xi}_t$. We then define three energy terms: the **RGB energy** $E_{\text{rgb}}(\mathcal{I}, \mathcal{G}) = \sum_t \mathcal{L}_{\text{MSE}}(\hat{\mathbf{I}}_t, \mathbf{I}_t) + \mathcal{L}_{\text{LPIPS}}(\hat{\mathbf{I}}_t, \mathbf{I}_t)$, where $\mathbf{I}_t$ is the ground-truth color image and the loss combines MSE and LPIPS losses [96]; the **mask energy** $E_{\text{mask}}(\mathcal{M}, \mathcal{G}) = \sum_t \texttt{CE}(\hat{\mathbf{M}}_t, \mathbf{M}_t)$, where $\texttt{CE}$ is cross-entropy and $\mathbf{M}_t$ is either a given labeled mask [64, 89] or one inferred via segmentation tracking [24]; and the **monocular geometry energy** $E_{\text{mono}}(\mathcal{D}, \mathcal{G}) = \sum_t \|\hat{\mathbf{N}}_t - \mathbf{N}_t\|_2^2 + \mathcal{L}_{\text{norm}}(\hat{\mathbf{D}}_t, \mathbf{D}_t)$, where $\mathbf{N}_t$ and $\mathbf{D}_t$ are monocular normal and depth priors and $\mathcal{L}_{\text{norm}}$ is the scale- and shift-invariant L2 loss [91].

**Regularization Terms:** Because videos only partially observe a 3D scene, optimizing observations alone cannot yield a complete, plausible, and physically valid reconstruction; we therefore impose generative, geometric, and physical priors as regularizers to enable fully interactive 3D scenes.

The **completeness energy** $E_{\text{comp}}$ encourages complete reconstruction of each object's shape despite the partial observations. Inspired by generative image-to-3D methods [39], for each object $i$ we synthesize virtual observations $\tilde{\mathcal{O}}_i = \{\tilde{\mathcal{I}}_i, \tilde{\mathcal{D}}_i, \tilde{\mathcal{M}}_i, \tilde{\mathcal{N}}_i\}$ by "shooting" it from multiple virtual viewpoints with a pretrained multi-view diffusion model Wonder3D [39]. *Unlike* the single object setting for most image-to-3D works, because our complex scenes often feature inter-object occlusions (e.g., a sofa covered by a blanket), we first inpaint occluded regions using LaMa [66] before generating these views. Given the synthesized observations, we define the completeness energy as

$$E_{\text{comp}} = \sum_i \big(E_{\text{mask}}(\tilde{\mathcal{I}}_i, \{\mathbf{v}_i\}) + E_{\text{rgb}}(\tilde{\mathcal{M}}_i, \{\mathbf{v}_i\}) + E_{\text{mono}}(\tilde{\mathcal{D}}_i, \{\mathbf{v}_i\})\big), \quad (3)$$

where $E_{\text{rgb}}, E_{\text{mono}}, E_{\text{mask}}$ are the observation losses defined similarly in our observation terms, although they are measured at virtual viewpoints here.

The **geometry energy** $E_{\text{geo}}$ ensures geometry compatibility between each object, such that their geometry does not intersect with each other:

$$E_{\text{geo}}(\mathcal{G}) = \sum_i \left(E_{\text{pene\_sdf}}(\mathbf{g}_i; \mathcal{G}) + E_{\text{pene\_mesh}}(\mathbf{g}_i; \mathcal{G})\right). \quad (4)$$

The SDF-penetration term $E_{\text{pene\_sdf}} = \sum_{\mathbf{x} \in R_i} \sum_{k \neq i} \max\big(0, -g_k(\mathbf{x}) - g_i(\mathbf{x})\big)$ ensures no two object SDFs overlap, where $R(i) = \{\mathbf{x} \in \mathbb{R}^3 | \arg\min_k g_k(\mathbf{x}) = i\}$ is the set of points belong to instance $i$. Intuitively, if $\mathbf{x}$ lies in instance $i$, then for any other instance $k$, $g_k(\mathbf{x}) \geq -g_i(\mathbf{x})$ must hold to prevent intersections. Similarly, each object's mesh should not intersect with any other object mesh. This can be measured by measuring whether intersecting two meshes resulting empty set or not: $E_{\text{pene\_mesh}} = \mathbf{1}(\texttt{inter}(\mathcal{M}_i, \mathcal{M}_j) \neq \emptyset)$.

Finally, it is important to ensure that our recovered digital twin of the scene is simulatable; hence, physical plausibility is crucial. To this end, we introduce **physics energy**, which measures physical plausibility via two terms:

$$E_{\text{physics}} = E_{\text{stable}} + E_{\text{touch}} = \texttt{Diff}\big(\mathcal{T}, \texttt{Sim}(\mathcal{T}, \mathbf{a}_{\text{gravity}}; \mathcal{G})\big) + \sum_{(i,j) \in \mathcal{E}} \texttt{dist}(\mathcal{M}_i, \mathcal{M}_j). \quad (5)$$

The stable term $E_{\text{stable}}(\mathcal{G}) = \texttt{Diff}\big(\mathcal{T}, \texttt{Sim}(\mathcal{T}, \mathbf{a}_{\text{gravity}}; \mathcal{G})\big)$ quantifies translational and rotational deviations of each object, with $\texttt{Diff}(\mathcal{T}, \mathcal{T}') = \sum_i(|\texttt{trans}(\mathbf{T}_i^{-1}\mathbf{T}_i')| + |\texttt{rad}(\mathbf{T}_i^{-1}\mathbf{T}_i')|)$ and $\texttt{Sim}$ is

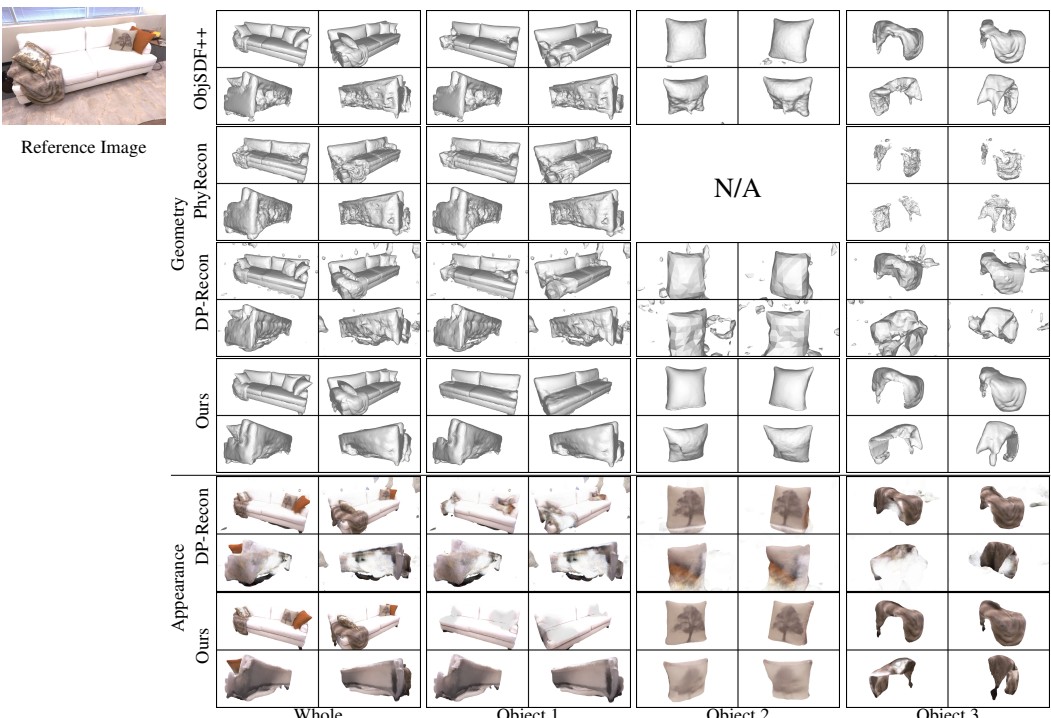

Figure 3: **Qualitative Comparisons on Object Geometry and Appearance Reconstruction:** Our method delivers superior reconstructions by smoothly inpainting occluded regions with LaMa and completing invisible back-facing geometry with Wonder3D. Unlike baselines, our approach eliminates object interpenetration, ensuring physical stability during simulation.

the forward physical simulator step as defined in Eq. 1; a low $E_{\text{stable}}$ indicates static equilibrium under gravity, i.e. scene remains static in the simulator. The touch term $E_{\text{touch}} = \sum_{(i,j)\in\mathcal{E}} \texttt{dist}(\mathcal{M}_i, \mathcal{M}_j)$ encourages each supporting pair $(i, j)$ to make contact, $\texttt{dist}$ is the Chamfer distance between meshes.

### 3.3 Inference

Optimizing the scene graph from Sec. 3.2 is challenging because it mixes discrete variables (graph topology, object–object relations) with continuous ones (neural SDFs, Gaussians, and physical parameters) and includes non-differentiable terms like physical stability. We therefore use a three-stage divide-and-conquer approach: first, recover initial geometry and appearance by minimizing the observation terms; next, refine shapes and physical parameters by minimizing the geometry and physics terms through generative sampling combined with structured tree search; and finally, fine-tune appearance by re-minimizing observation terms. This yields a fully plausible, interactive 3D scene (Fig. 2).

**Stage 1: Gradient-based Optimization:** We first optimize each object node's appearance $\mathbf{a}_i$ and geometry $g_i$ to match the observations $\mathcal{O}$ via gradient-based optimization. Specifically, we minimize the observation terms plus SDF-penetration regularization through differentiable volume rendering—similar to neural SDF methods [75, 74, 30]—to obtain per-instance SDFs $g_i$. Additionally, we recover small objects by balancing training samples across all instances. We then extract initial meshes $\mathcal{M}_i$ via marching cubes and refine each object's Gaussians $\mathbf{f}_i$ via splat rendering and RGB rendering, mask, and monocular geometry losses, yielding our dual scene representation per each instance [16].

**Stage 2: Sampling-based Optimization:** The Stage 1 scene model supports freeview rendering and accurate visible-region geometry but remains incomplete, non–physical, and non–interactive. Directly minimizing $E_{\text{complete}}$, $E_{\text{physics}}$, and $E_{\text{geo}}$, however, is challenging due to complex high-order interactions (e.g., multi-object physical interaction), intrinsic multi-modality (invisible regions admit multiple solutions), and non-differentiable components (e.g., mesh intersections, physics simulations). To address this, we adopt an approach that combines the diverse proposal capability of

**generative sampling** with the combinatorial optimization strength of **tree search** to minimize our structured objective.

*Scene graph edges creation:* After the gradient-based optimization, our framework infers the topology of the scene graph $\mathcal{G}$ from the instance meshes extracted from neural SDF, where edges $\mathcal{E}$ encode support relations in a tree rooted at $\mathbf{v}_0$ (the background, e.g., the room). We build this tree by analyzing physical contact and support relationships between objects: each object is assigned a parent based on which surface provides its primary support, as determined by the geometry and orientation of contact surfaces. We begin by identifying objects resting directly on the background, then recursively process the remaining objects in an order that respects the physical dependency structure—objects that support others are registered before those they support. Starting from $\mathbf{v}_0$, we repeat until all observed instances have been added to the tree.

*Generative sampling:* We begin by sampling diverse, complete shapes for each instance: we prompt Wonder3D's multi-diffusion model with real-world observations and generate virtual views $\tilde{\mathcal{I}}_i$ from various viewpoints. Thanks to its generative nature, resampling multiple times with different seeds yields diverse virtual observations even from the same viewpoints. Then, for each virtual observation, we minimize $E_{\text{comp}}$ independently, producing a diverse set of 3D shape candidates per object.

*Structured tree search:* We have generated multiple complete shape samples per instance that all fit the observations, but it remains unclear which combination is most physically plausible. Exhaustively evaluating every combination is impractical, since the physical-plausibility energy $E_{\text{physics}}$ entails a high-order combinatorial optimization. To address this, we perform a tree search over our generative samples. Starting at the root node, we traverse each node in breadth-first order; at each active node (object), we evaluate $E_{\text{physics}}$ for all samples and retain the sample with the lowest energy among those evaluated. We then adjust its state and physical parameters to enforce stability and prevent interpenetration (see details in the supplementary material).

*Remark:* The key novelty and advantage of the proposed inference algorithm is the combination of generative sampling with a structured tree search for amodal, physically plausible reconstruction. Unlike scene-level amodal methods [10] that rely on asset retrieval, our sampling is asset-free and generates input consistent, diverse shape hypotheses. Unlike prior simulation-verification methods [49, 48], which only enforces object–ground consistency, our tree search ensures global stability along every support chain. By driving a non-differentiable simulator (e.g., IsaacSim) end-to-end, we eliminate any reconstruction-to-deployment gap.

**Stage 3: Gradient-based Refinement**    Since Stage 2 adjusts object states, physical parameters, and shape, it is necessary to further refine the Gaussians attached to the surface to ensure a complete and realistic appearance. To this end, we fine-tune Gaussians for all objects using splat rendering by minimizing the observation terms via gradient descent. This yields our final scene graph.

## 4   Experiments

**Dataset:**    We conduct the experiments across multiple datasets: 3 scenes from Replica [64], 3 scenes from Scannet++ [89], 2 scenes from iGibson [26], and one self-captured scene. The Replica and Scannet++ datasets cover diverse indoor structures and lighting conditions, and the iGibson dataset offers complete geometry of every object, allowing per-object reconstruction evaluation. Instance masks are provided by dataset annotations or estimated with SAM [24].

**Metrics:**    We evaluate geometry quality with Chamfer Distance (CD), F-Score (F1), and Normal Consistency (NC) [91], and assess rendering quality using PSNR, SSIM, and LPIPS. For physical evaluation, we adopt consistent physical parameters, put all scene components in the Isaac Sim [47], and measure stability with translation/rotation changes when gravity is applied. The stability ratio is calculated as: Stable $\% = \frac{\#\text{Stable Instances}}{\#\text{All Instances}}$, where instances are identified as stable if changes are under a certain threshold. We also report the object reconstruction ratio: OR$\% = \frac{\#\text{Reconstructed Instances}}{\#\text{All Instances}}$, which quantifies how many objects are present in the final reconstruction, regardless of their completeness.

**Baselines:**    We evaluate our framework against SOTA approaches in instance-aware amodal 3D scene reconstruction. **ObjectSDF++** [75] uses per-instance SDF for scene representation. **PhyRecon** [48] extends instance-aware scene reconstruction by incorporating a differentiable physical loss to

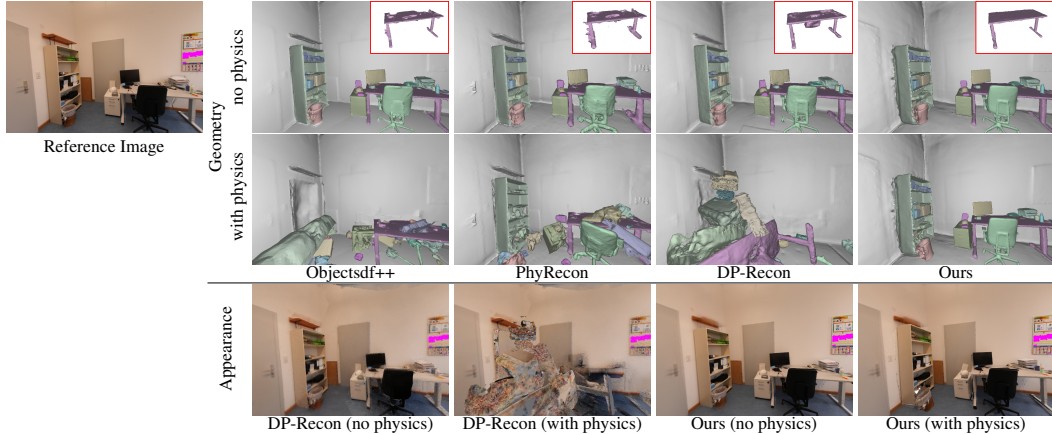

Figure 4: **Qualitative Comparisons on Physical Simulation:** We compare geometry layouts and appearance before and after physical simulation, with the table geometry reconstructions highlighted in inset figures. HoloScene's complete, non-interpenetrating geometry remains stable in physics simulators, unlike baseline methods. Our Gaussian on mesh delivers high-quality, real-time rendering throughout the simulation process.

| | | Method | Geometry | | | Rendering | | | Real Time | Physics | | |
|---|---|---|---|---|---|---|---|---|---|---|---|---|
| | | | CD↓ | F1↑ | NC↑ | PSNR↑ | SSIM↑ | LPIPS↓ | | OR%↑ | Stable (Ground)%↑ | Stable (All)%↑ |
| Scene | Replica | ObjSDF++ | 6.72 | 64.36 | 88.53 | 29.12 | 0.851 | 0.355 | ✗ | 98.6 | 78.3 | 39.4 |
| | | PhyRecon | 4.52 | 71.07 | 92.06 | 23.19 | 0.764 | 0.434 | ✗ | 77.5 | 56.5 | 5.6 |
| | | DP-Recon | 3.45 | 87.66 | 94.23 | 22.10 | 0.728 | 0.420 | ✓ | 56.3 | 21.7 | 8.5 |
| | | Ours | 4.05 | 83.21 | 92.21 | 27.82 | 0.849 | 0.304 | ✓ | 100.0 | 95.7 | 81.7 |
| | Scannet++ | ObjSDF++ | 25.20 | 70.71 | 87.15 | 27.46 | 0.887 | 0.292 | ✗ | 96.5 | 81.6 | 28.2 |
| | | PhyRecon | 31.16 | 39.57 | 82.28 | 22.32 | 0.791 | 0.432 | ✗ | 92.9 | 67.3 | 9.4 |
| | | DP-Recon | 22.96 | 65.48 | 87.13 | 21.44 | 0.715 | 0.466 | ✓ | 90.6 | 20.0 | 9.4 |
| | | Ours | 21.93 | 63.11 | 88.09 | 25.88 | 0.873 | 0.268 | ✓ | 100.0 | 93.9 | 70.6 |
| | iGibson | ObjSDF++ | 12.33 | 38.64 | 83.74 | 29.60 | 0.891 | 0.299 | ✗ | 65.0 | 44.2 | 36.1 |
| | | PhyRecon | 11.27 | 45.49 | 83.85 | 27.40 | 0.860 | 0.333 | ✗ | 62.9 | 45.3 | 5.2 |
| | | DP-Recon | 30.31 | 21.89 | 70.81 | 21.94 | 0.728 | 0.432 | ✓ | 74.2 | 16.3 | 4.1 |
| | | Ours | 12.00 | 34.15 | 82.91 | 25.88 | 0.854 | 0.301 | ✓ | 100.0 | 74.4 | 71.1 |
| Object | iGibson | ObjSDF++ | 3.52 | 79.03 | 75.30 | 11.03 | 0.571 | 0.134 | ✗ | 65.0 | 44.2 | 36.1 |
| | | PhyRecon | 5.47 | 70.71 | 71.89 | 8.92 | 0.609 | 0.250 | ✗ | 62.9 | 45.3 | 5.2 |
| | | DP-Recon | 5.81 | 61.31 | 70.61 | 13.90 | 0.770 | 0.301 | ✓ | 74.2 | 16.3 | 4.1 |
| | | Ours | 3.17 | 81.31 | 78.13 | 16.55 | 0.863 | 0.185 | ✓ | 100.0 | 74.4 | 71.1 |

Table 2: **Quantitative Results on Scene Reconstruction:** HoloScene's generative sampling and scene graph-based tree search produce the most physically plausible reconstructions while preserving high-quality geometry and supporting real-time and realistic rendering. We highlight the best and second-best methods with distinct colors. For rendering quality, we only compare with DP-Recon's texture mesh-based rendering since ObjSDF++ and PhyRecon lack real-time rendering capabilities.

optimize unstable objects. However, it only handles object–ground interactions and does not support hierarchical or inter–object relationships. **DP-Recon** [49] incorporates diffusion Score Distillation Sampling (SDS) [55] for amodal sparse-view reconstruction. However, it does not handle inter-object occlusions. All three methods use instance-level SDFs, so the SDF values of different instances might interfere with one another. We adopt their open-source codes and adapt them for the testing benchmarks. Please refer to the supplementary material for more implementation details.

**Implementation details:** Our inference pipeline consists of three stages. Stage 1 employs gradient-based optimization for 100k steps with loss weights $\lambda_{rgb} = 1.0$, $\lambda_{mask} = 0.5$, $\lambda_{depth} = 0.5$, and $\lambda_{normal} = 0.1$. Stage 2 uses sampling-based optimization with $\lambda_{rgb} = 2.0$, $\lambda_{mask} = 0.5$, $\lambda_{depth} = 10.0$, $\lambda_{normal} = 10.0$, $\lambda_{pene} = 5.0$, generating three samples per instance. Stage 3 refines Gaussians via gradient-based optimization with $\lambda_{t1} = 0.95$ and $\lambda_{ssim} = 0.05$. The optimization takes approximately 4 hours, 4 hours, and 20 minutes for stages 1, 2, and 3, respectively, on a single A6000 GPU. We utilize Marigold [14] for monocular depth and normal estimation.

| Level | Dataset | TexMesh / GoM | Physical Energy | Scene Graph | CD↓ | F1↑ | NC↑ | PSNR↑ | SSIM↑ | LPIPS↓ | OR%↑ | Stable (All) %↑ |
|-------|---------|---------------|-----------------|-------------|-----|-----|-----|-------|-------|--------|------|-----------------|
| Scene | Replica | TexMesh | ✗ | ✗ | 3.50 | 88.00 | 92.30 | 26.45 | 0.810 | 0.329 | 100.0 | 43.7 |
|       |         | GoM | ✗ | ✗ | 3.50 | 88.00 | 92.30 | 28.30 | 0.845 | 0.349 | 100.0 | 43.7 |
|       |         | GoM | ✓ | ✗ | 4.32 | 80.19 | 92.10 | 27.14 | 0.839 | 0.334 | 100.0 | 69.0 |
|       |         | GoM | ✓ | ✓ | 4.05 | 83.21 | 92.21 | 27.82 | 0.849 | 0.304 | 100.0 | 81.7 |
| Object | iGibson | GoM | ✗ | ✗ | 4.20 | 79.22 | 76.96 | 13.74 | 0.735 | 0.204 | 100.0 | 41.2 |
|        |         | GoM | ✓ | ✓ | 3.17 | 81.31 | 78.13 | 16.55 | 0.863 | 0.185 | 100.0 | 71.1 |

Table 3: **Ablation Study on Model Design:** We observe improved physical stability and object-level reconstruction with the help of generative priors, physical energy, and scene graph representation, and there is a trade-off between scene-level reconstruction and instance-level physical stability.

## 4.1 Experimental Results

**Scene-level evaluations:** We first evaluate the reconstruction at the scene level. In Tab. 2, we compare our framework with the three baselines across Replica, Scannet++. and iGibson dataset, with qualitative results shown in Fig. 3 and Fig. 4. Our method achieves the best physical reconstruction results in terms of both object reconstruction ratio and stable object ratio, while maintaining comparable scene-level reconstruction quality in geometry and appearance. Unlike baselines [75, 48, 50] where small objects often disappear due to SDF interference from larger adjacent objects, our framework benefits from balancing training samples across all instances during the optimization stage 1, recovering all instances in the scenes. PhyRecon assumes that all objects rest directly on the ground, which damages geometry when objects are supported by other objects. DP-Recon prioritizes completeness via SDS over physical stability, failing to adequately address object interpenetration. In contrast, our sampling-based optimization and completion approach yields the most physically stable results.

**Object-level evaluations:** We evaluate object-level reconstruction using the iGibson dataset, which provides complete ground truth geometry and appearance for each object. This evaluation is especially challenging as it tests the reconstruction of occluded regions that models never observe. Our evaluation compares reconstructed geometry directly with ground truth and renders 6 viewpoints around each object to assess appearance quality. Results in Tab. 2 demonstrate that our method outperforms all baselines in reconstructing invisible and occluded regions, validating our framework's effectiveness in completing objects beyond directly observed surfaces.

**Ablation study:** Our ablation study (Tab. 3) reveals the contribution of each component. Switching from textured mesh to Gaussian rendering improves visual quality, while adding physics energy with generative priors enhances physical stability. The scene graph inter-object relationships further improve physics performance by better reconstructing occluded regions, especially those from `support` relationships. However, we identify a trade-off between scene-level reconstruction accuracy and physical stability, where optimizing for physical plausibility may occasionally compromise pixel-alignment with original observations.

**Failure rate analysis:** We conduct a statistical analysis of failure modes in geometry and physics reconstruction. In Tab. 4, we compare HoloScene against reconstruction-based baselines [75, 48, 50] across geometry failures on iGibson and physics failures on all three datasets, with failure criteria defined by F-score thresholds (70.0) and simulation stability metrics. Geometry failures include invalid surfaces, partial and oversized shapes, while physics failures arise from invalid shapes incompatible with simulators, overlooked object-object contacts, inter-object penetration causing repelling forces, and inaccurate contact modeling leading to positional drift. Our method achieves the lowest failure rates in both categories. Unlike ObjSDF++ [75] and PhyRecon [48] which lack shape priors and produce invalid geometry, our generative priors eliminate all invalid geometry failures through reliable structural guidance and balanced sampling. DP-Recon [50] emphasizes visual completeness over physical constraints, resulting in high penetration failures. Additionally, PhyRecon's ground-contact assumption overlooks contact pairs that can destabilize the simulation. In contrast, we integrate generative priors with physics-aware constraints directly into the reconstruction process, which substantially reduces both invalid shapes and inter-object penetrations, demonstrating the effectiveness of physical consistency in the optimization.

| Method | Geometry Failure (iGibson) | | | | | Physics Failure Rate (%) | | | Physics Failure Breakdown (All Datasets) | | | | |
|---|---|---|---|---|---|---|---|---|---|---|---|---|---|
| | Total # | Fail # | Invalid # | Partial # | Oversized # | Replica | ScanNet++ | iGibson | Total # | Invalid # | Overlap # | Penetration # | Drifting # |
| ObjSDF++ | 97 | 52 | 34 | 13 | 5 | 60.6 | 71.8 | 53.6 | 235 | 38 | 0 | 76 | 52 |
| PhyRecon | 97 | 64 | 36 | 21 | 7 | 94.4 | 90.6 | 66.0 | 235 | 58 | 121 | 33 | 24 |
| DP-Recon | 97 | 70 | 25 | 33 | 12 | 91.5 | 90.6 | 72.2 | 235 | 64 | 0 | 102 | 69 |
| Ours | 97 | **24** | **0** | **13** | 11 | **18.3** | **29.4** | **24.7** | 235 | **0** | **0** | **6** | 60 |

Table 4: **Quantitative Failure Rate Comparison:** HoloScene achieves significantly lower geometry and physics failure rates compared to reconstruction-based baselines, demonstrating more robust 3D reconstruction and physically plausible scene modeling.

## 4.2 Interactive Environment Applications

**Real-Time Interactive Game** With our reconstructed environment, we can create a real-time interactive game with Unreal Engine [12]. As Fig. 1 shows, we build a third-person game with the reconstructed texture meshes. The objects could be physically rearranged in the game world, and the game agent could also interact with the scene through realistic physics.

**Interactive 3D Editing** In our simulation environment, we could also achieve high-quality interactive 3D editing by moving the object Gaussians with its underlying physical mesh geometry. In Fig. 1, we demonstrate this by changing the location and orientation of the interactable chair.

**Immersive Experience Recording** We show our interactable reconstructed 3D objects with immersive experience recording. Given a static RGB video of a person manipulating an object, we aim to recover the object's 6D pose and resimulate its motion in a virtual 3D scene. We recover the camera pose with VGGT [69], adjust the predicted depth [54] to align with the virtual scene, and adopt FoundationPose [73] for object tracking with our reconstructed 3D object for model-based 6D pose estimation. As shown in Fig. 1, we enable consistent replay of real-world interactions in virtual scenes while accurately recovering the object's pose from visual input.

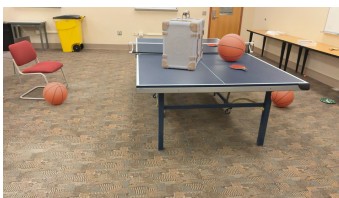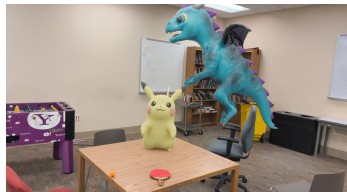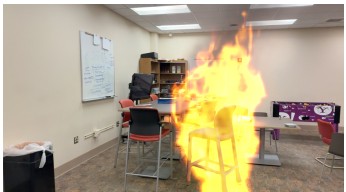

Figure 5: **Dynamic VFX Results.** We augment the inferred interactive 3D scene with various visual effects such as dropping objects, adding animations, and fires.

To enhance immersion, we augment the scene with dynamic visual effects, including rigid body simulations, character animations, and particle effects. We adopt visual effects from AutoVFX [18] to overlay virtual content and shadows onto the image. As Fig. 4 shows, we produce effects that blend naturally with the scene.

## 5 Conclusion & Limitation

We presented HoloScene, a novel interactive 3D modeling framework that uses scene graphs and energy-based optimization to reconstruct environments with realistic appearance, complete geometry, and interactive physical plausibility, achieving superior real-time rendering, geometric accuracy, and stable simulation. **Limitations**: HoloScene currently only handles videos of static indoor scenes; dynamic scenes and large outdoor environments remain challenging. Future work will focus on relightable reconstruction and extending support to articulated and deformable objects.

**Acknowledgment:** This project is supported by the Intel AI SRS gift and NSF Awards #2331878, #2340254, #2312102, #2414227, and #2404385. We greatly appreciate the NCSA for providing computing resources through Delta and Delta AI program.

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

# Appendix

## A   More Interactive Environment Application Results

**Real-Time Interactive Game**    With the reconstruction results of HoloScene, we could create an interactive game in Unreal Engine [12], which supports real-time interactions with our created assets. We import all the objects and the background texture mesh into the game engine and create a third-person game, where the agent can freely explore, navigate, jump, and interact with the objects in the scene. Thanks to the complete geometry and appearance, as well as the great physical plausibility of HoloScene, the objects could remain stable in the game. Also, the movements of objects could follow physical rules when the objects interact. The Fig. 1 shows the game screenshots, and more videos could be found in the supp. material as well.

**Interactive 3D Editing**    With each object's appearance and geometry reconstructed, we support interactive 3D scene editing by allowing users to manipulate individual Gaussians through a control panel. Objects can be translated along the 2D xy-plane and rotated around the vertical (z) axis. As illustrated in Fig. 2, the reconstructed swivel chair is repositioned freely and rendered realistically, resulting in coherent and intuitive scene modifications.

**Immersive Experience Recording**    We demonstrate the usage of the reconstructed 3D objects from HoloScene for immersive experience recording. Given a static RGB video of a person manipulating an object, our goal is to recover the object's 6D pose and resimulate its motion within a virtual 3D scene. We use the first frame of the video as the target view for camera calibration. To determine the camera pose, we first estimate relative transformations between the target and nearby training views using VGGT [69]. We then solve a least-squares problem with Powell's method [56] to derive a global transformation that aligns the VGGT-predicted poses with the ground-truth poses of the training views. This transformation is applied to register the target view within the training coordinate system. We use UniDepthV2 [54] to predict the depth map for each frame and compute a scale ratio between the predicted monocular depth and depth rendered from the 3D Gaussians. This ratio adjusts the predicted depth to align accurately within the virtual scene. For object tracking, we adopt FoundationPose [73], using our high-quality reconstructed 3D asset as a CAD model for model-based 6D pose estimation. The recovered pose sequence is then applied to the 3D object and its 3D Gaussians for final rendering. As illustrated in Fig. 3, we accurately recover the object's pose from visual input, enabling consistent replay of real-world interactions in virtual scenes.

**Dynamic Visual Effects**    We showcase an immersive application by augmenting the reconstructed scenes with dynamic visual effects. These include rigid body simulations, 3D character animations, and particle-based effects, all spatially grounded within the virtual scene. We adopt multiple visual effects created by AutoVFX [18] to simulate virtual content and overlay it, along with shadows, onto the input image. As shown in Fig. 4, we produce effects that blend naturally with the scene, enhancing both visual richness and physical realism.

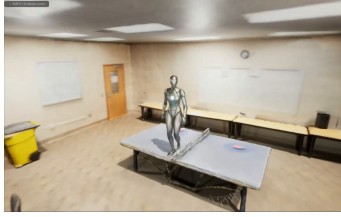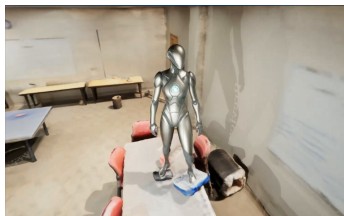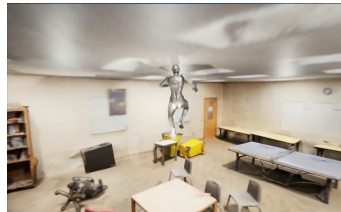

Figure 1: **More Interactive Game Results:** We show more screenshots of the interactive game built by Unreal Engine here. The agent can run, jump, and interact with all the objects in the game.

## B   More Physical Simulation Results

In this section, we will show more physical simulation results of Holoscene and its comparison with the baseline methods [75, 48, 49]. In Fig. 5 and Fig. 6, we show the comparisons of our method and

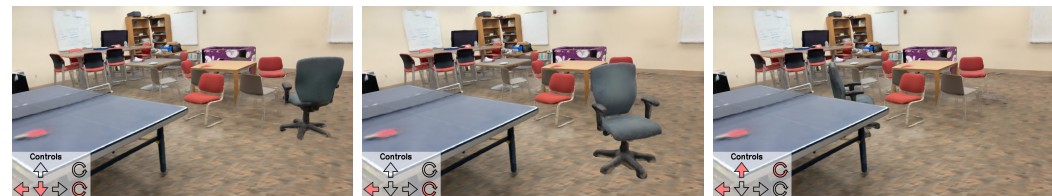

Figure 2: **More Interactive 3D Editing Results:** The user could interact with the object in the scene, change the location and orientation of the object the control panel, and attain the edited scene.

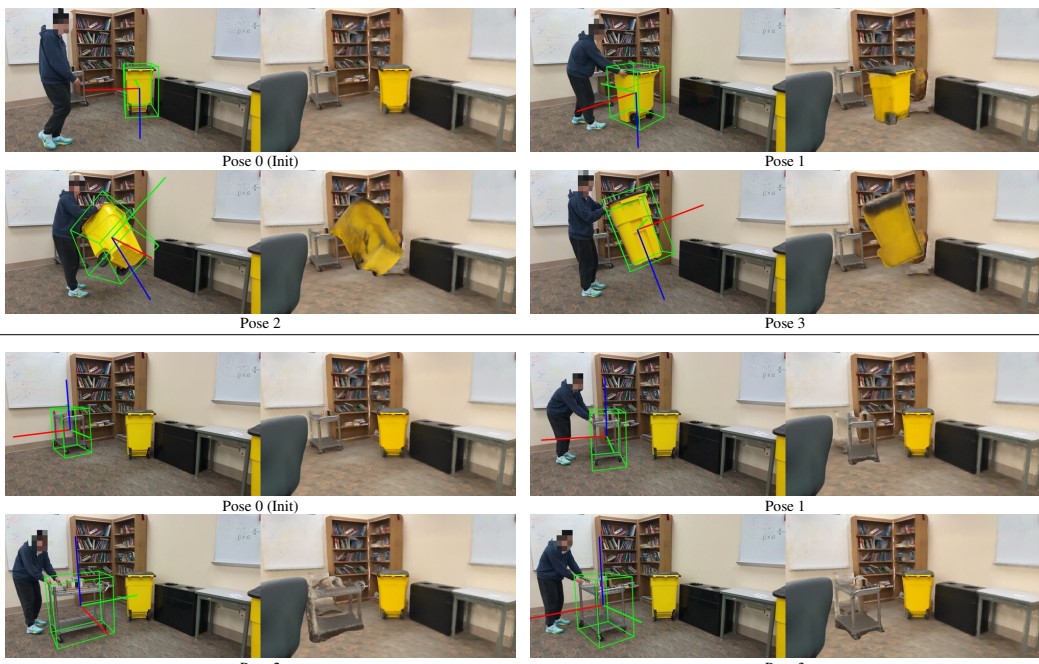

Figure 3: **More Immersive Experience Recording Results:** Here we show more results of the immersive experience recording of two objects in the scene. For each image pair, the original recording is shown on the left, and the simulated object rendering with Gaussians is on the right. The estimated object 6D poses are marked with oriented bounding boxes.

the baselines in the Scannet++ [89] dataset. In Fig. 7, we show the comparisons in the Replica [64] dataset. In Fig. 8, we show the comparisons in the iGibson [26] dataset. From the results, we can observe that HoloScene can achieve the best physical stability and plausibility, as well as realistic rendering quality.

## C Method Details

**Structured Tree Search** When we perform the breadth-first traversal on the scene tree structure, for each node, we will end up choosing the reconstructed object candidate with the lowest physical energy. Then we will further adjust the state of the chosen candidate object to ensure its stability. Specifically, we will regularize the SDF value and the mesh vertex locations to prevent the interpenetration of objects. Before applying marching cube to the instance SDF, we will first prune the SDF value of the points if those points fall into the boundary of its parent object or the nearby objects. After we attain the mesh with marching cube, we will perform a collision test of the reconstructed object mesh with its parent object and the nearby objects. And if there exist collisions, we will dynamically optimize the locations of the vertices to avoid the collisions. After the optimization, the contact surfaces of

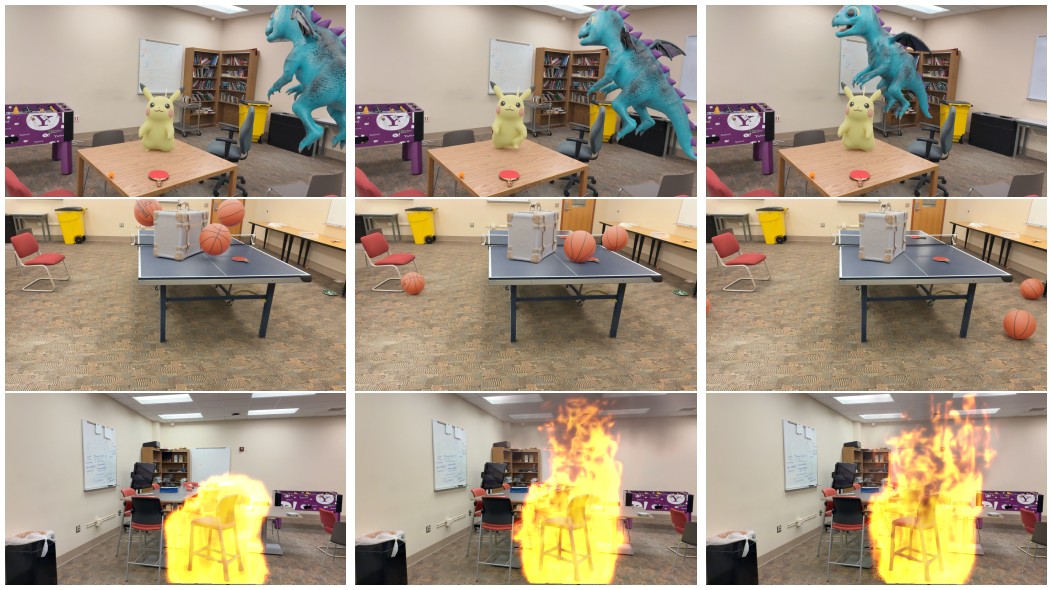

Figure 4: **More Dynamic Visual Effect Results:** We show more results of dynamic visual effects, including animations, dropping objects, and setting fires.

objects will be regularized and avoid inter-object collisions, so that the stability of the scene can be further improved.

## D  Experiment Details

### D.1  Datasets

We adopt three datasets as we mentioned in our main paper: Replica [64], Scannet++ [89], and iGibson [26]. **Replica:** The original Replica dataset only provides the texture mesh with instance annotations, and no existing videos with camera trajectories are provided. We manually design a dense camera trajectory of several hundred frames for each scene, and render images with a size of (512, 512). **Scannet++:** The scannet++ dataset provides multi-view posed images and the aligned 3D scan with instance annotation. However, limited by the quality of the scan, the instance mask derived from the scan annotation is not ideal. We adopt [24] to get cleaner instance masks instead. For the image resolution, we downsample the original image by 2. **iGibson:** The iGibson dataset provides the complete geometry and appearance of every object and the background scene, but without provided camera trajectories. We manually design a dense camera trajectory of several hundred frames for each scene, and render images with a size of (512, 512).

### D.2  Baselines

We adopt the open-source codes of the baselines and adapt them for our benchmark. Our baselines are ObjectSDF++ [75], PhyRecon [48], and DP-Recon [49]. Now we will elaborate the details of them.

**ObjectSDF++:**  The original codebase only supports their rendering images from Replica [64], where each image has a size of (384, 384). In order to make it adaptable to other image resolutions, we implement a new data loader in the code.

**PhyRecon:**  Similar to ObjectSDF, the original code base only supports their rendering images from Replica [64] and a different resolution of resized images from Scannet++ [89]. For the adaption to other image resolutions, we implement a new data loader in the code. The PhyRecon has a final stage

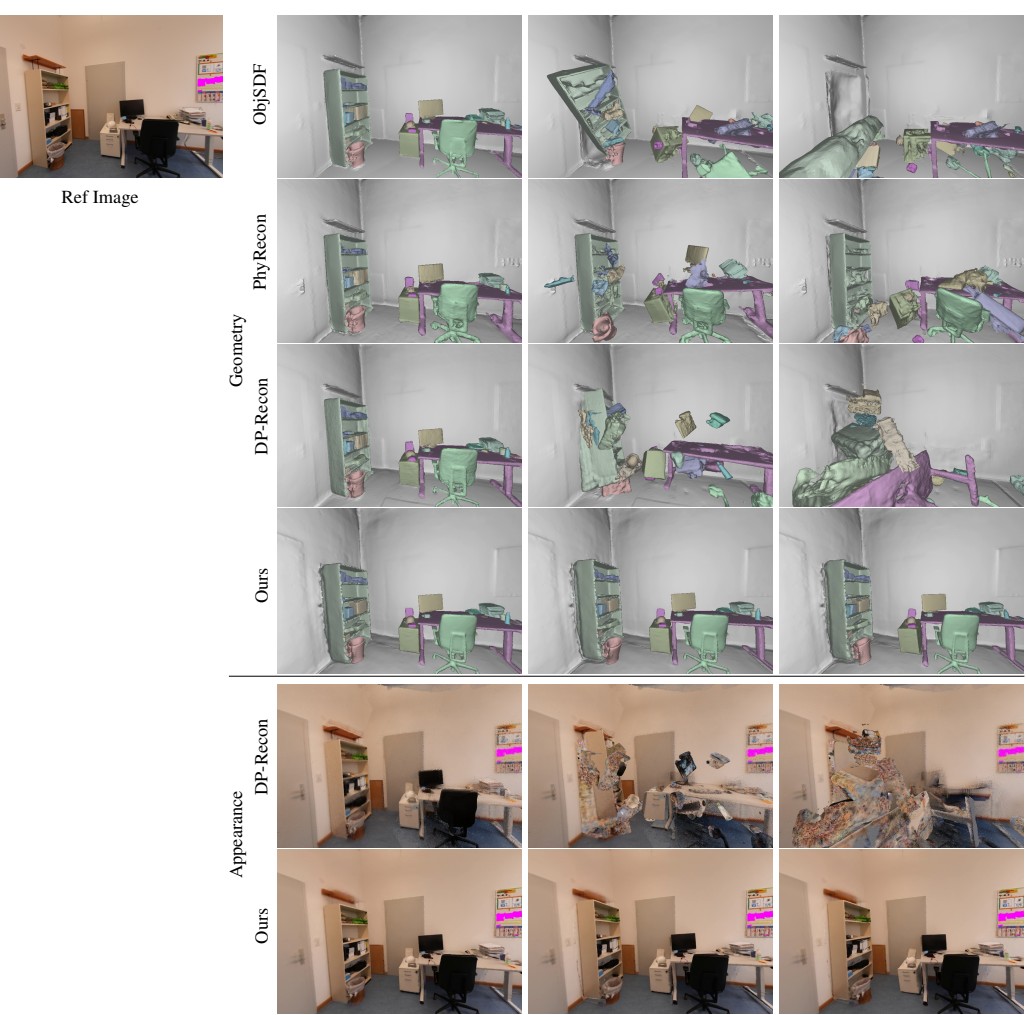

Figure 5: **More Physical Simulation Results in Scannet++ Dataset Scene 1:** We show the geometry and appearance of each object in the scene with physical simulation, from no physics state, intermediate state with physics, and final state with physics.

of applying physical loss to every object. But it may happen that some objects fail to reconstruct themselves, so the physical loss can not be applied to those objects. In this case, we skip those objects.

**DP-Recon:**  Similar to the baselines above, we implement a new data loader to adapt the code to our dataset. DP-Recon has the geometry stage and the following appearance stage. We follow the officially released code to run those experiments. DP-Recon needs to generate the object bounding boxes in the running process. But DP-Recon overlooks the possibility that some objects may fail to be reconstructed and disappear. So in this case, we set the bounding box of the object as the largest possible bounding box in the scene to fix the issue. We evaluate the appearance, geometry, and physics based on the final texture mesh results from the second stage of DP-Recon.

### D.3  The inference details of HoloScene

The inference stage of HoloScene is composed of three stages after generating the scene graph in stage 0, as we mentioned in the main paper. In stage 1 of the gradient-based optimization, we follow the previous instance-level SDF [75] to set the hyperparameters and train for 200k iterations similarly. In stage 2 of the sampling-based optimization, we adopt the Isaac Sim [47] as the physical simulator,

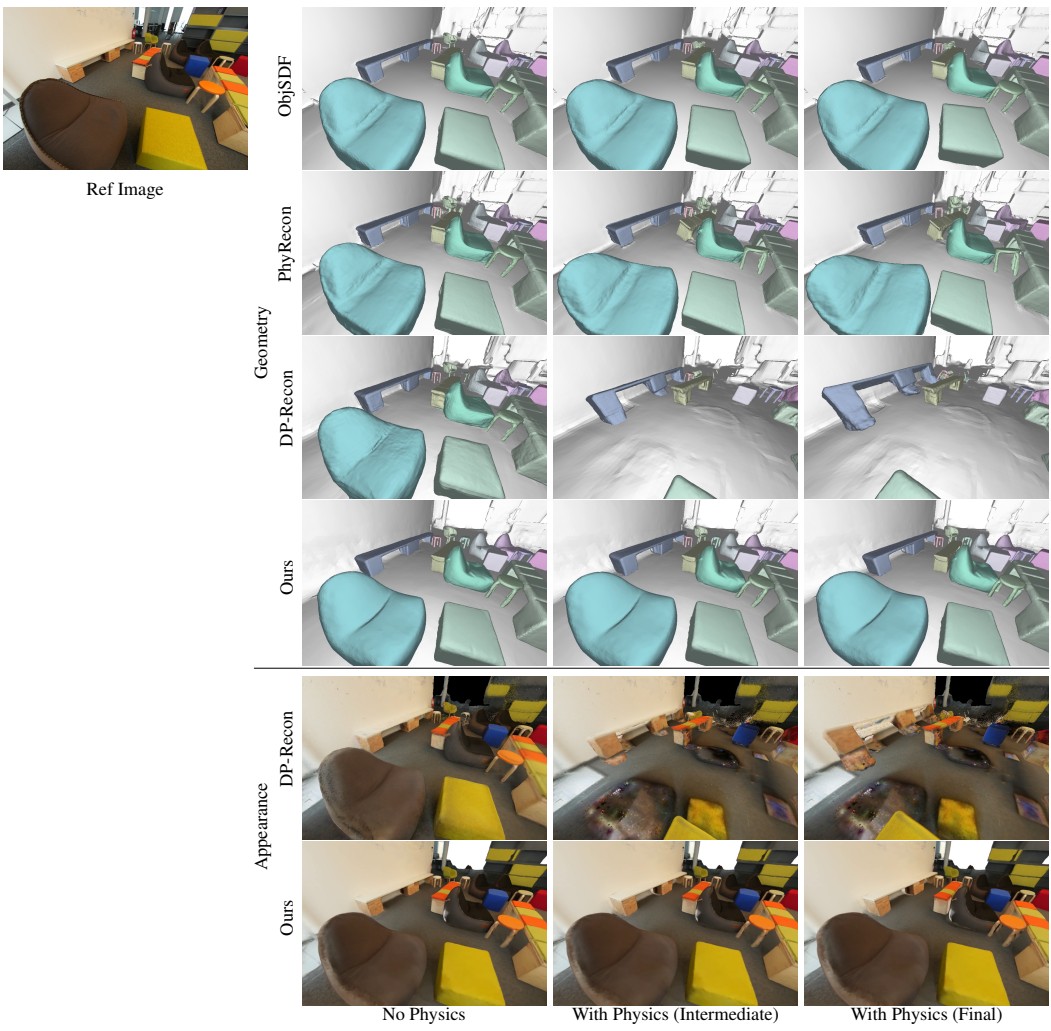

Figure 6: **More Physical Simulation Results in Scannet++ Dataset Scene 2:** We show the geometry and appearance of each object in the scene with physical simulation, from no physics state, intermediate state with physics, and final state with physics.

and set the physical parameters in the simulation as default. In the sampling with generative models, we randomly choose the seeds and generate at most five candidates for each object. In stage 3 of the gradient-based refinement, we refine the trained Gaussians globally. We optimize the Gaussians for 30k iterations following [23], and in each iteration, we also use generated images to optimize the instance Gaussians as well. For other learning parameters, they are set to default in our experiments.

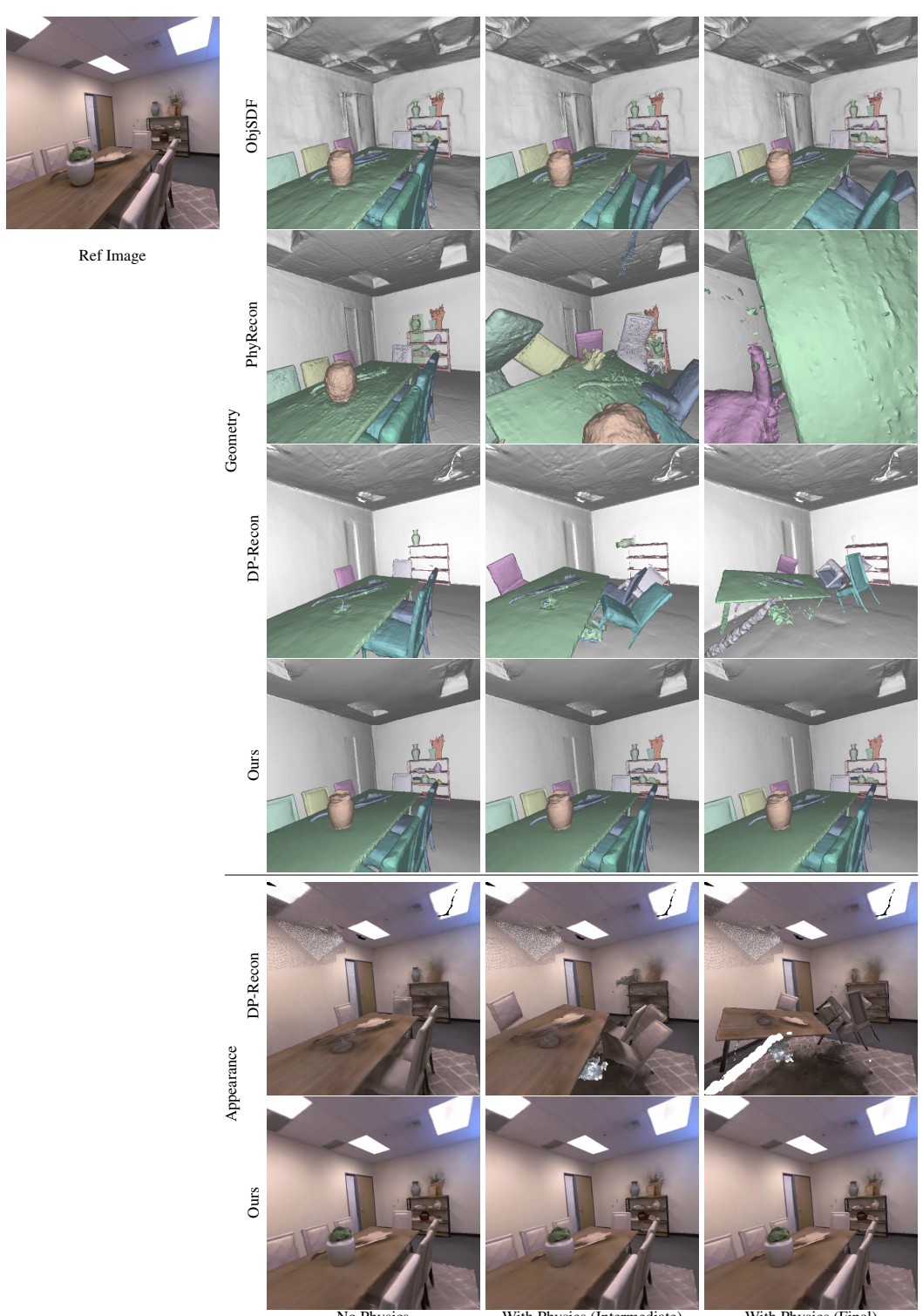

Figure 7: **More Physical Simulation Results in Replica Dataset:** We show the geometry and appearance of each object in the scene with physical simulation, from no physics state, intermediate state with physics, and final state with physics.

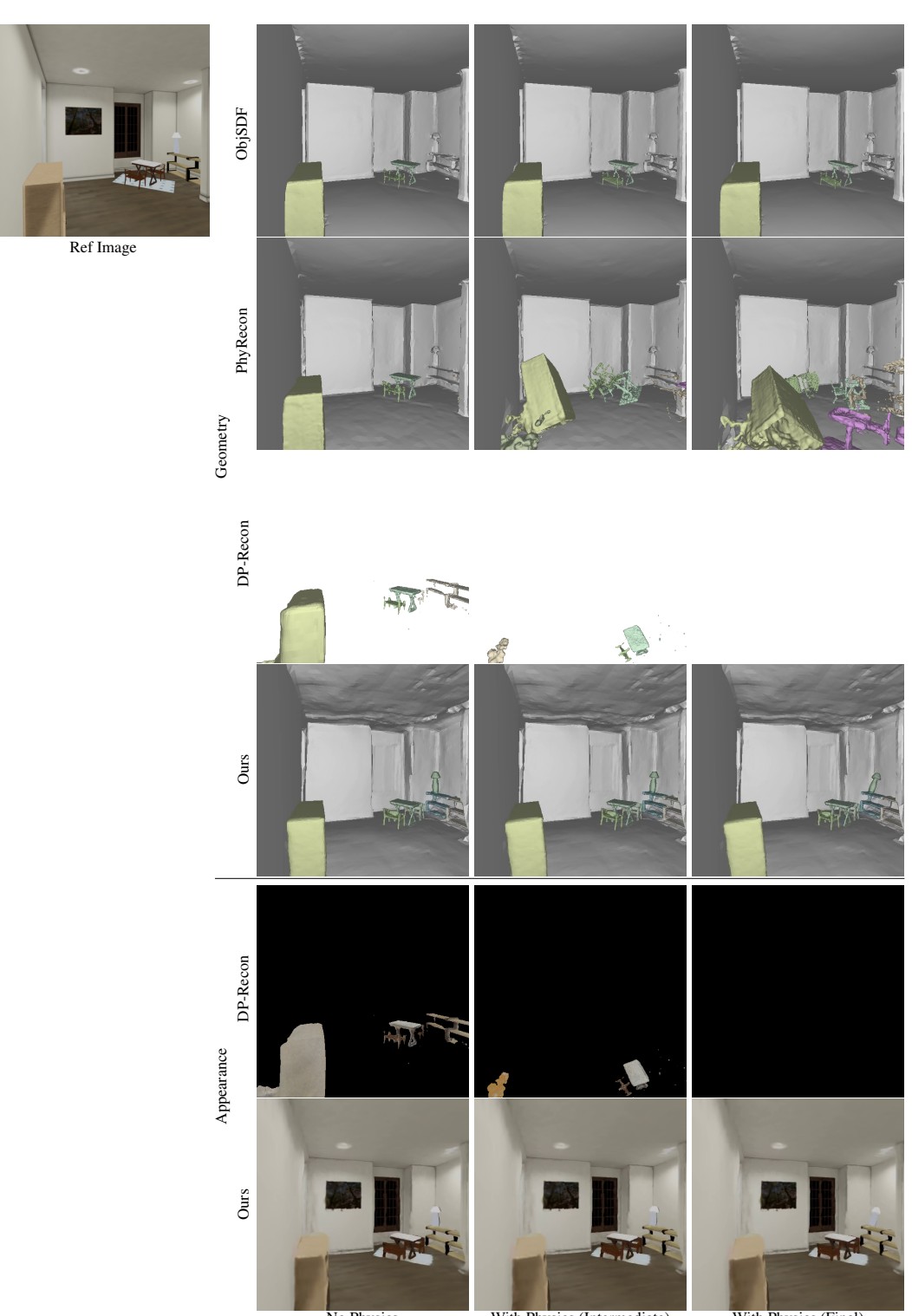

Figure 8: **More Physical Simulation Results in iGibson Dataset:** We show the geometry and appearance of each object in the scene with physical simulation, from no physics state, intermediate state with physics, and final state with physics. **DP-Recon fails to reconstruct the complete background mesh, leading to object falling in the figure.**

