# OpenReview forum: "HoloScene: Simulation‑Ready Interactive 3D Worlds from a Single Video"
_NeurIPS.cc/2025/Conference — NeurIPS 2025 poster_

### Official Review · Reviewer_BhAa · 2025-06-11

**Clarity:** 2
**Significance:** 3
**Originality:** 2
**Rating:** 4
**Confidence:** 4

**Summary:**

This paper presents HoloScene, which aims to create an interactive, simulation-ready digital scene of an environment from a single input video. The main steps of the pipeline are to: (1) construct a scene graph hierarchy by recursively querying a VLM, (2) fit the 3d representation (parameterized by SDFs and 3D gaussians) via NeRF/3DGS-style optimization, (3) complete "unseen" properties (parts of objects not seen in input images + physics properties) by sampling generative priors (Wonder3D + LAMA) and a physics simulator (IsaacSim), (4) a final round of NeRF/3DGS-style finetuning to improve visual appearance. The resulting representation allows interactive 3D edition and the overlay of VFX effects (ie: set an asset on fire).

The authors compare HoloScene to prior work across indoor scenes from iGibson, Replica, and ScanNet++. Qualitatively, the physical interactions seem much more plausible and the scene geometry in unobserved regions looks a bit better. The supplement contains some nice illustrations of possible downstream applications, such as using the resulting environment in an interactive game.

**Questions:**

- Table 2: I understand that ObjSDF++ is not real-time, but we are still seeing a 2-4db PSNR gap that we don't typically see when comparing 3DGS to implicit representations. It's also really confusing how PSNR and SSIM/LPIPS don't really seem correlated, especially for the iGibson results where all of the PSNR values are very low. Could the authors provide a more detailed analysis of how to interpret these results? If accepted, I'd strongly recommend that the extra page in camera-ready version be used to provide some illustrative qualitative examples that illustrate these discrepancies.
- Line 217-218: "Thanks to its generative nature, resampling multiple times with different seeds yields diverse virtual observations even from the same viewpoints" - I was confused about this statement - what would you want different observations from the same viewpoints? I would have assumed that multiview consistency is very important for 3D reconstruction, so I suspect that I am misunderstanding this statement.
- Could the authors confirm that they commit to releasing the training code upon acceptance? This would greatly alleviate my reproducibility concerns and I'd likely adjust my score accordingly.

**Ethical Concerns:**

["NO or VERY MINOR ethics concerns only"]

**Final Justification:**

Thank you for answering my questions - most of my concerns have been addressed and I will maintain my positive rating.

**Limitations:**

Yes

**Quality:**

3

**Strengths And Weaknesses:**

Strengths:

- The visual results, although not 100% photorealistic at times, are impressive. I very much appreciate the videos the authors share in the supplement!
- The overall design clearly requires quite a bit of thought, and the use of both gradient-based optimization + more discrete sampling to effectively use external priors seems like a sensible approach

Weaknesses:
- The pipeline involves a lot of discrete components (VLM, diffusion model, monocular depth prior, physics simulator, etc), and if any of the components behave incorrectly these errors might compound and the overall pipeline might degrade significantly. The authors note that the pipeline only works for indoor scenes at the moment but it would be interesting to get a sense of how the pipeline works in more challenging / less "in-distribution" scenes. Even if performance is not great, that would still help give us a sense of the approach's generalizability.
- A major concern I have with this paper is that it omits a lot of training details that will make it very difficult to reproduce considering the complexity of the pipeline. A non-exhaustive list of missing important details include: loss hyperparameters, number of gradient optimization and sampling steps, average training time, what VLM/monocular depth models that are being used, and how the SDF penetration term is computed efficiently (a naive implementation seems prohibitively expensive). The authors mention additional details in the supplement, but I do not see any in the attached zip. In fairness, the authors committed to releasing code in the questionnaire, which would alleviate my concerns if they honor this commitment, but I would still like to see more details in the paper
- The exposition of the writing could be improved - in particular I think that the authors should be more clearly state upfront what parts of the pipeline are novel contributions vs which have been inspired by prior work. For example, using a scene graph was originally proposed by Neural Scene Graphs (Ost el at, CVPR 2021), which should be explicitly cited and it's unclear if the use of a VLM to generate the graph is a novel contribution or something also proposed by other work.
- The use of color highlighting in Table 2 is misleading / the results are hard to interpret (see questions section)


Typos:
- Physical plausible -> physically plausible

---

> ### Author Rebuttal · Authors · 2025-07-30
>
> We thank the reviewer for appreciating our framework design and recognizing our videos as impressive results.
>
> We now address the concerns.
>
> **Robustness of HoloScene:**
> HoloScene exhibits robustness to individual foundation model failures by jointly optimizing multiple energy terms and selecting the best among sampled solutions. Failures in one module typically introduce inconsistencies, which are reflected as higher energy, making such samples less likely to be selected. This contrasts with traditional multi-stage pipelines, where a single module failure can propagate downstream and compromise the entire output. This robustness is reflected in both our real-world examples—where foundation models are imperfect—and in benchmark metrics.
>
> That said, this sampling-based robustness is not a guarantee against all types of failures. Catastrophic errors or insufficient sampling coverage, as shown in some of our failure cases, can still lead to imperfect final results.
>
> **Outdoor scenes:**
> HoloScene, like other compared methods, such as DP-Recon[1] and ObjSDF++[2], mainly focuses on and conducts experiments on indoor scenes. Outdoor scene setting features dynamic motions and unbounded geometry, which is out of our claimed scope. We will leave the outdoor simulation-ready environment reconstruction for future work.
>
> **Training details:**
> We will open-source our full codebase upon acceptance to support reproducibility. For the reviewer's reference, we provide additional training details in the table below and will include these parameters in the final version.
>
> *Gradient-based optimization (stage 1):*
> | Parameter | $\lambda_{rgb}$ ($\mathrm{E}_{obs}$) | $\lambda_{mask}$ ($\mathrm{E}_{obs}$) | $\lambda_{depth}$ ($\mathrm{E}_{obs}$) | $\lambda_{normal}$ ($\mathrm{E}_{obs}$) | total_steps |
> |:---------:|:------------------------------------:|:-------------------------------------:|:--------------------------------------:|:---------------------------------------:|:-----------:|
> |   Value   |                 1.0                  |                  0.5                  |                  0.5                   |                   0.1                   |    100k     |
>
> *Sampling-based optimization (stage 2):*
> | Parameter | $\lambda_{rgb}$ ($\mathrm{E}_{comp}$) | $\lambda_{mask}$ ($\mathrm{E}_{comp}$) | $\lambda_{depth}$ ($\mathrm{E}_{comp}$) | $\lambda_{normal}$ ($\mathrm{E}_{comp}$) | $\lambda_{pene}$ ($\mathrm{E}_{pene}$) | sampling # per instance |
> |:---------:|:-------------------------------------:|:--------------------------------------:|:---------------------------------------:|:----------------------------------------:|:--------------------------------------:|:-----------------------:|
> |   Value   |                  2.0                  |                  0.5                   |                  10.0                   |                   10.0                   |                  5.0                   |            3            |
>
> *Gradient-based Refinement (stage 3):* (for Gaussian refinements)
>
>
> | Parameter | $\lambda_{l1}$ | $\lambda_{ssim}$ |
> |:---------:|:--------------:|:----------------:|
> |   Value   |      0.95      |       0.05       |
>
> *Average Training time (one A6000 GPU):*
>
> | Stage |  0  |  1  |  2  |  3  |
> |:-----:|:---:|:---:|:---:|:---:|
> | Time  | 10m | 4h  | 4h  | 20m |
>
> *Models used:*
>
>
> | Model Type |  VLM   | monocular cues |
> |:----------:|:------:|:--------------:|
> | Model Name | GPT-4o |    Marigold[3]    |
>
> **SDF penetration term details:**
> In each iteration, we sample 4096 points uniformly in the 3D space, and calculate the SDF values of all instances at those points. We then apply $E_{\mathrm{pene\_sdf}} = \sum_{\mathbf{x} \in R_i}\sum_{k\neq i}\max\bigl(0,\,-g_k(\mathbf{x}) - g_{i}(\mathbf{x})\bigr)$ on these points as the penetration loss to optimize. By uniformly sampling points, we can reduce the calculation complexity.
>
> **Scene graph inference, writing, and novelty:**
> Thank you for the suggestion. We will expand the related work section to include a discussion of prior work on scene graphs, including Neural Scene Graphs [Ost et al., CVPR 2021], and cite relevant references.
>
> We do not claim the use of a VLM for scene graph inference as our main contribution. Rather, we emphasize that the **scene graph plays a critical role** in HoloScene for enforcing physically plausible interactions (see Tab. 3, L280–L286). While the idea of a scene graph itself is not novel, our **energy-based formulation and sampling-based inference framework built on top of the scene graph** is a core contribution of HoloScene. This integrated approach enables robust and physically consistent scene reconstruction from video.
>
> **Evaluation results analysis:**
> Here we explain our evaluation results in Tab. 2. We will include a more detailed explanation in the final version.
>
> - *Real-time PSNR gap:* We adopt the Gaussian on Mesh (GoM) from DRAWER[4]. Compared with standard 3DGS, the position, orientation, scale, and other properties of every Gaussian are regularized, so they can align better with mesh surfaces. While this strategy limits the rendering quality, the real-time Gaussian rendering aligns with the accurate geometry, improving visual quality in downstream simulation applications. This is a key trade-off for attaining a simulation-ready environment. It is noticeable that the gap happens in DRAWER[4], GaMeS[5], and other GoM papers as well.
>
> - *Correlation of PSNR/SSIM/LPIPS:* PSNR and SSIM/LPIPS are correlated when we compare the real-time rendering quality in Tab. 2. But those three metrics sometimes are not necessarily correlated. For example, we can witness uncorrelated PSNR and SSIM/LPIPS in Table 1 of the Mip-NeRF 360[6] paper.
>
> - *iGibson results:* The object-level evaluation on iGibson scenes shows low PSNR values because the evaluation is performed on the whole object, including unseen regions, making it especially challenging. (L274-L275) Therefore, the PSNR/SSIM/LPIPS are lower than common reconstruction task.  We will add more qualitative examples in the camera-ready version.
>
> **Diverse results of resampling:**
> Thank you for pointing this out — there may have been a misunderstanding. By "diverse virtual observations," we refer to the diversity in *invisible* regions when generative resampling with different seeds. This diversity helps us explore multiple plausible completions and select the one that best minimizes our regularization terms.
>
> Importantly, multi-view consistency is preserved through the use of an explicit 3D representation (GoM), ensuring that renderings from different viewpoints remain consistent with each other.
>
> **Code open source:**
> We will open-source all the code, data, and running scripts upon acceptance, and we believe this will encourage future research from the community.
>
> [1] Ni, Junfeng, et al. "Decompositional neural scene reconstruction with generative diffusion prior." Proceedings of the Computer Vision and Pattern Recognition Conference. 2025.
>
> [2] Wu, Qianyi, et al. "Objectsdf++: Improved object-compositional neural implicit surfaces." Proceedings of the IEEE/CVF International Conference on Computer Vision. 2023.
>
> [3] Ke, Bingxin, et al. "Repurposing diffusion-based image generators for monocular depth estimation." Proceedings of the IEEE/CVF conference on computer vision and pattern recognition. 2024.
>
> [4] Xia, Hongchi, et al. "Drawer: Digital reconstruction and articulation with environment realism." Proceedings of the Computer Vision and Pattern Recognition Conference. 2025.
>
> [5] Waczyńska, Joanna, et al. "Games: Mesh-based adapting and modification of gaussian splatting." arXiv preprint arXiv:2402.01459 (2024).
>
> [6] Barron, Jonathan T., et al. "Mip-nerf 360: Unbounded anti-aliased neural radiance fields." Proceedings of the IEEE/CVF conference on computer vision and pattern recognition. 2022.

---

> > ### Comment · Reviewer_BhAa · 2025-08-06
> >
> > Thank you for answering my questions. Although the rebuttal addresses some of my questions, I echo x1eu's concern method's reliability and robustness, and would also be interested in the comparisons they are suggesting.

---

> > > ### Author Response · Authors · 2025-08-08
> > > **(1/2) Response to Reviewer BhAa's Follow-Up Comments**
> > >
> > > Thank you for your valuable feedback. Regarding the reliability and robustness, we responded to reviewer x1eu with additional experiments and statistic analysis as suggested. We paste our analysis here in this thread.
> > >
> > >
> > > ## **Quantitative Failure Rate Comparison**
> > >
> > > Following your suggestion, we provide a `statistical analysis of HoloScene’s failure rate` and a `direct comparison with current reconstruction-based SOTA` methods that `aim to achieve similar goals`, including ObjSDF++, PhyRecon, and DP-Recon.
> > >
> > > We quantify both *geometry failures* and *physics failures*—the most common failure modes in simulation-ready asset reconstruction—and identify typical causes for each. This provides a clear root-cause analysis of where and how our method improves over prior work, highlighting the reliability and robustness of our framework.
> > >
> > > ### **Geometry failure**
> > >
> > > **Definition**. For this study, a predicted geometry is considered failed if its F-Score (F1) is below 70, using a 0.05 m threshold compared to the GT geometry.
> > >
> > > **Reason and Phenomenon**. Geometry failures are mainly caused by:
> > > * *Invalid object surfaces* — surface shape is null, non-manifold or non-watertight, often due to noisy observations, poor optimization, or insufficient regularization, leading to invalid signed distance fields.
> > > * *Partial shape* — caused by the absence of shape priors (e.g., ObjectSDF++ and PhyRecon) or imperfect priors (e.g., DP-Recon, HoloScene).
> > > * *Oversized shape* — typically resulting from imperfect priors or under-optimized physical constraints.
> > >
> > > **Evaluation**. We evaluate geometry failures on the iGibson sub-dataset. The table below summarizes the the results.
> > >
> > > |  Method   | Total objects # | Failure # &darr; | Failure rate % &darr; | Invalid geometry # | Partial shape # | Oversized shape # |
> > > |:---------:|:---------------:|:----------------:|:---------------------:|:---------------:|:------------------:|:--------------------:|
> > > | ObjSDF++  |       97        |        52        |         53.6          |       34        |         13         |          **5**           |
> > > | PhyRecon  |       97        |        64        |         66.0          |       36        |         21         |          7          |
> > > | DP-Recon  |       97        |        70        |         72.2          |       25        |         33         |          12          |
> > > | HoloScene |       97        |      **24**      |       **24.7**        |        **0**        |         **13**         |          11          |
> > >
> > >
> > > The results reveal several key insights:
> > > - **Generative priors reduce invalid geometry**:
> > >   Methods using generative shape priors (DP-Recon, HoloScene) significantly reduce *invalid geometry* cases. Unlike ObjSDF++ and PhyRecon, which lack such priors, DP-Recon uses a SDS-based loss that helps but still sometimes produces invalid shapes due to optimization challenges. In contrast, HoloScene achieves **zero invalid cases**, demonstrating high reliability through balanced sampling across all instances as well as generative inpainting and sampling.
> > > - **Sampling improves shape accuracy**:
> > >   Despite using SDS loss, DP-Recon performs worse on *partial* and *oversized* shapes than ObjSDF++ and PhyRecon, likely due to the limitations of SDS in completing shapes. HoloScene leverages sampling-based generative inference combined with physical constraints and energy-based scoring. While it doesn’t eliminate all size errors, it offers a strong overall performance.

---

> > > > ### Author Response · Authors · 2025-08-08
> > > > **(2/2) Response to Reviewer BhAa's Follow-Up Comments**
> > > >
> > > > ### **(Continue)**
> > > >
> > > > ---
> > > >
> > > > ### **Physics failure**
> > > >
> > > > **Definition**. The reconstructed object has a movement over 0.05 of the scene unit length or rotates over 10 degrees in the simulator after applying gravity in the composed scene (the same threshold as we evaluate stability in paper).
> > > >
> > > > **Reason and Phenomenon**. Physics failures are mainly caused by:
> > > >
> > > > * *Invalid shapes* — Missing or incomplete surfaces make objects incompatible with the physics simulator.
> > > > * *Overlooked contact pairs* — Some methods (e.g., PhyRecon) focus only on object-ground contacts (as stated in Supp D.1: `focused solely on object-ground support for simplicity and training efficiency`), ignoring object-object interactions. This weakens physical supervision (PhyRecon Sec. 3.2).
> > > > * *Penetration* — Inter-object geometry penetration causes repelling forces during simulation, leading to instability.
> > > > * *Drifting* — Inaccurate shape modeling at contact points (even with minimal penetration) can cause gradual drifting from the original position.
> > > >
> > > >
> > > > **Evaluation**. We evaluate physics failures across **all datasets** used in the paper: Replica, ScanNet++, and iGibson. The table below summarizes the `physics failure rate` for each dataset.
> > > >
> > > > |  Method   | Replica % &darr; | Scannet++ % &darr; | iGibson % &darr; |
> > > > |:---------:|:----------------:|:------------------:|:----------------:|
> > > > | ObjSDF++  |       60.6       |        71.8        |       53.6       |
> > > > | PhyRecon  |       94.4       |        90.6        |       66.0       |
> > > > | DP-Recon  |       91.5       |        90.6        |       72.2       |
> > > > | HoloScene |     **18.3**     |      **29.4**      |     **24.7**     |
> > > >
> > > > To better understand the sources of failure, we also break down the failure counts by underlying cause:
> > > >
> > > > |  Method   | Total object # | Failure # &darr; | Failure rate % &darr; | Invalid shapes # | Overlooked contact pairs # | Penetration # | Drifting # |
> > > > |:---------:|:--------------:|:----------------:|:---------------------:|:-------:|:-------------------:|:-----------:|:--------:|
> > > > | ObjSDF++  |      253       |       166        |         65.6          |   38    |          0          |     76      |    52    |
> > > > | PhyRecon  |      253       |       236        |         93.3          |   58    |         121         |     33      |    24    |
> > > > | DP-Recon  |      253       |       235        |         92.9          |   64    |          0          |     102     |    69    |
> > > > | HoloScene |      253       |      **66**      |       **26.1**        |    0    |          0          |      6      |    60    |
> > > >
> > > > The results reveal several key insights into physics failure across methods:
> > > > - **Failure Tied to Physics Modeling**:
> > > >   ObjSDF++ and DP-Recon often fail due to *invalid shapes* and *penetrations*, stemming from limited or no physics modeling. PhyRecon includes physics but suffers from high failure rates by considering only object-ground contact, neglecting object-object interactions (PhyRecon Supp D.1).
> > > > - **Shape Priors vs. Physical Plausibility**:
> > > >   DP-Recon emphasizes visual completeness with shape priors but lacks physical stability, leading to drifting and penetration. PhyRecon ensures ground contact but doesn't generalize to more complex interactions.
> > > > - **HoloScene’s Integrated Solution**:
> > > >   HoloScene combines *generative priors*, *scene graph reasoning*, and a *simulator-as-critic* to model object-object contact and enforce physical plausibility. This reduces penetration failures and eliminates invalid or oversimplified geometry.
> > > >
> > > > These findings suggest that physical consistency should not be treated as an afterthought—whether through *post-hoc correction* or *surrogate loss terms*—but instead integrated directly into the generative reconstruction process via *hard constraints* and *physical validation*. Joint reasoning over structure, geometry, and physics emerges as a promising direction for building more robust, simulation-ready digital assets. We thank the reviewer again for the nice suggestion.

---

### Official Review · Reviewer_Np4v · 2025-06-22

**Clarity:** 3
**Significance:** 3
**Originality:** 1
**Rating:** 2
**Confidence:** 4

**Summary:**

This paper proposes HoloScene, a system for reconstructing physically plausible and interactive 3D scenes from monocular video. It models the scene as a structured scene graph with object-level geometry, appearance, and physics attributes. The method combines gradient-based and sampling-based optimization, and uses simulation feedback to ensure physical stability. The reconstructed scenes support editing, rendering, and deployment in interactive environments.

**Questions:**

1.You mention in the paper that your method follows the Gaussians-on-Mesh design from DRAWER. However, DRAWER is not included in the experimental comparisons. Given the similarity in task setting and overall pipeline, could you provide a detailed comparison with DRAWER, both quantitatively and qualitatively?

2.Your method also shares many core components with CAST, including relation graph modeling, object-centric decomposition, and generative geometry completion. Could you clarify what you consider to be the key methodological differences and novel contributions beyond these prior works?

3.More generally, please elaborate on the originality of your approach. What specific design choices or technical insights distinguish your method from existing systems like CAST and DRAWER, beyond the use of multiview input?

If you can address my concerns, I will increase my score.

**Ethical Concerns:**

["NO or VERY MINOR ethics concerns only"]

**Final Justification:**

I appreciate the authors' rebuttal and acknowledge the engineering effort involved. However, I believe the proposed method remains largely an incremental extension of prior work like CAST, with limited methodological novelty. While technically sound, the contribution does not meet the originality bar expected for NeurIPS.

**Limitations:**

yes

**Quality:**

3

**Strengths And Weaknesses:**

Strengths：

1.The system is well-engineered and demonstrates solid integration of geometry, appearance, and physics.

2.Results are visually compelling, and the method supports downstream applications such as interaction and simulation.

Weaknesses：

1.The method can be seen as a **multiview** extension of CAST[1], sharing similar core components such as relation graph modeling, object-centric geometry generation, structure-aware decomposition, and physics-based reasoning. I find the level of originality to be limited.

2.Closely related works like DRAWER [2] and CAST are cited but not included in experiments, despite having **highly similar goals** and **setups**, making it difficult to assess whether the observed improvements come from the proposed method or from different input settings.

3.The critique of CAST in the related work section is also questionable. For example, HoloScene highlights CAST’s reliance on generative models, but it similarly uses Wonder3D for multiview geometry synthesis.

Overall, the contribution lies more in system integration and engineering effort than in novel methodology.

[1]CAST: Component-Aligned 3D Scene Reconstruction from an RGB Image

[2]DRAWER: Digital Reconstruction and Articulation With Environment Realism

---

> ### Author Rebuttal · Authors · 2025-07-30
>
> We thank the reviewer for appreciating HoloScene as a solid integrated system with visually compelling results and various downstream applications.
>
> We now address the concerns.
>
> **Comparison with CAST [1]:**
> HoloScene differs from CAST in problem setting, inference stage, and overall purpose. We provide a detailed comparison in Table 1 and the related work section, and summarize key differences below:
>
> * **Problem setting:**
> CAST generates *component-aligned* 3D scenes from a single *image*, and its outputs do not necessarily match the input image in appearance or geometry (see CAST’s overview figure). In contrast, HoloScene reconstructs *simulation-ready digital twins* from a single *video*, aiming to replicate geometry, appearance, and physical plausibility from the input video.
>
> * **Inference stage:**
> CAST relies on a generative model and uses differentiable optimization refining objects to avoid object collisions. HoloScene combines *reconstruction*, *generation* with *sampling-based optimization*, minimizing both observation energy (to match input video appearance) and regularization energy (for physical plausibility and object completion). While it leverages Wonder3D [3] for generation, reconstruction is guided by observed data.
>
> * **Purpose:**
> CAST focuses on content creation with clean, component-aligned 3D assets from single images. HoloScene instead targets realistic, faithful, physically consistent reconstruction from video for simulation-ready environments. Both realism and fine-level fidelity are our key goals.
>
> **Comparison with DRAWER [2]:**
> We discussed DRAWER[2] in the related work section (L69-L71). Though HoloScene adapts Gaussians on Mesh from DRAWER[2], we have distinct differences with them. Here we will explain the detailed comparisons of HoloScene and DRAWER[2]. We also include a table below for the reviewer's reference.
> * **Focus:**
> DRAWER[2] focuses on the *articulated* object reconstruction and lacks the shape completion and stable physics for some *rigid-body* objects. However, HoloScene focuses on *all* *rigid-body* objects in the input video, and reconstructs them with invisible region completion, collision avoidance, and physical stability in the simulation.
> * **Approach:**
> In DRAWER, *rigid-body* objects are decomposed following Video2Game[4]. From the paragraph of *Rigid object decomposition* in DRAWER's supp material page 9, we can see that, although *rigid-body* objects in DRAWER can be segmented from the SDF field using *3D bounding box*, DRAWER doesn't decompose every rigid-body objects and the objects lack the completion for invisible regions, so that they can not be guaranteed with physical stability in the simulation. Unlike DRAWER[2], HoloScene decomposes every *rigid-body* object in the video. Instead of direct segmentation from the SDF field, every object is modeled by its own SDF for convenient 3D decomposition. HoloScene also adapts a generative sampling approach to optimize the completion and stability of every rigid-body object. We show our quantitative comparison results with DRAWER in the table below. We compare on all the rigid-body objects in the scene.
> * **Experiments:**
> We compare the scene-level and object-level performance on the iGibson dataset with DRAWER[2]. We provide the ground-truth object 3D bounding boxes to DRAWER. We show the evaluation results in the table below. From the results, we can witness that DRAWER[2] fails to get complete geometry (according to object-level CD/F1/NC and PSNR/SSIM/LPIPS evaluation) and plausible physics from the input video, and can not ensure stability even for objects on the ground (according to the stability evaluation).
>
> |  Method   | Visual Input | Real-time Rendering | Twin Fidelity | Articulated Object 3D completion | Rigid-body Object 3D completion | Holistic Decomposition | Physics Capacity | Physics Optimization |
> |:---------:|:------------:|:-------------------:|:-------------:|:--------------------------------:|:-------------------------------:|:----------------------:|:----------------:|:--------------------:|
> |  DRAWER   |    video     |      &#x2713;       |   &#x2713;    |             &#x2713;             |            &#x2718;             |        &#x2718;        |     &#x2718;     |       &#x2718;       |
> | HoloScene |    video     |      &#x2713;       |   &#x2713;    |             &#x2718;             |            &#x2713;             |        &#x2713;        |      Scene       |   Diff & Sampling    |
>
>
> |  Method   | Level  | CD &darr; | F1 &uarr; | NC  &uarr; | PSNR &uarr; | SSIM  &uarr; | LPIPS &darr; | Stable (Ground) % &uarr; | Stable (All) % &uarr; |
> |:---------:|:------:|:---------:|:---------:|:----------:|:-----------:|:------------:|:------------:|:------------------------:|:---------------------:|
> |  DRAWER   | Scene  |   11.06   |   43.22   |   84.36    |    26.03    |    0.863     |    0.278     |           41.9           |         40.2          |
> | HoloScene | Scene  |   12.00   |   34.15   |   82.91    |    25.88    |    0.854     |    0.301     |           **74.4**           |         **71.1**          |
> |  DRAWER   | Object |   3.53    |   79.27   |   76.73    |    15.72    |    0.843     |    0.201     |           41.9           |         40.2          |
> | HoloScene | Object |   **3.17**    |   **81.31**   |   **78.13**    |    **16.55**    |    **0.863**     |    **0.185**     |           **74.4**           |         **71.1**          |
>
> **Technical contribution and system design novelty:**
> We agree that system integration is a key part of HoloScene’s technical contribution. However, the novelty and intellectual contribution go beyond integration. Our energy formulation and inference design are both novel, enabling HoloScene to jointly optimize for appearance-geometry alignment and physical-geometric plausibility. This combination makes HoloScene an effective and principled framework for simulation-ready scene reconstruction.
>
>
> [1] Yao, Kaixin, et al. "Cast: Component-aligned 3d scene reconstruction from an rgb image." arXiv preprint arXiv:2502.12894 (2025).
>
> [2] Xia, Hongchi, et al. "Drawer: Digital reconstruction and articulation with environment realism." Proceedings of the Computer Vision and Pattern Recognition Conference. 2025.
>
> [3] Long, Xiaoxiao, et al. "Wonder3d: Single image to 3d using cross-domain diffusion." Proceedings of the IEEE/CVF conference on computer vision and pattern recognition. 2024.
>
> [4] Xia, Hongchi, et al. "Video2game: Real-time interactive realistic and browser-compatible environment from a single video." Proceedings of the IEEE/CVF Conference on Computer Vision and Pattern Recognition. 2024.

---

> > ### Comment · Reviewer_Np4v · 2025-08-04
> >
> > The authors claim that HoloScene differs from CAST in problem setting, inference stage, and overall purpose. However, these differences are mostly extensions in terms of **input modality and system-level integration**. For example, a different problem setting (single image vs. single video) does not, by itself, constitute methodological innovation.
> >
> > The core technical pipeline and key components are highly similar to those of CAST: object-centric generative geometry completion, VLM-based relation graph inference, SDF/geometry penetration constraints, and relation-based physical consistency optimization. HoloScene’s “sampling + tree search” essentially scores and selects among multiple generated candidates using the same energy terms, which is analogous to CAST’s generative alignment for handling multi-modality. This represents only a minor strategic variation rather than a new paradigm. Therefore, I still consider the originality to be limited, as the method is better characterized as a **“multi-view version of CAST with some engineering integrations.”**
> >
> > The authors argue that, unlike DRAWER, HoloScene completes “every rigid-body object” to ensure stability. However, DRAWER already includes geometry completion for rigid objects as part of its pipeline, so this does not constitute a substantial methodological difference. Regarding the reported comparison, it focuses only on rigid-body geometry and rendering metrics, omitting DRAWER’s primary evaluation axes such as articulation accuracy and motion plausibility, which are central to its problem setting. This raises concerns about the fairness and scope alignment of the comparison.
> >
> > I encourage the authors to provide a more comprehensive and balanced comparison, clearly articulating the methodological differences and including DRAWER’s primary evaluation metrics to ensure a fair and complete assessment.

---

> > > ### Author Response · Authors · 2025-08-05
> > > **(1/2) Response to Reviewer Np4v's Follow-Up Comments**
> > >
> > > We thank the reviewer for the constructive feedback.
> > >
> > > # Comparison with CAST
> > >
> > > **HoloScene differs significantly from CAST in terms of its** **problem setting**, **objective formulation**, **inference process**, and **emergent properties**.
> > >
> > > HoloScene’s objective formulation incorporates **explicit rendering and geometric consistency** via inverse neural rendering, as well as **physical stability** through simulator-in-the-loop optimization—features not present in CAST. To support this, we employ **generative sampling to directly optimize a unified energy formulation** (Eq. 2), in contrast to CAST’s **multi-step modular pipeline** (see Sec. 4, 5 in CAST). Together, this energy-based formulation and inference strategy form the basis of our **core methodological contributions**. These contributions are the result of thoughtful design decisions—not simply a consequence of extending to a multi-view setting.
> > >
> > > We therefore respectfully disagree with the reviewer’s characterization. HoloScene is **not merely** `a different problem setting (single image vs. single video),` nor is it just `a multi-view version of CAST with some engineering integration.`
> > >
> > >
> > > ## **Methodological Contribution Compared to CAST:**
> > >
> > > ### **I. Physical stability: CAST &#x2718;; HoloScene &#x2713;**
> > >
> > > CAST's *physics-aware correction* (Sec. 5 in CAST) only relies on its contact and support constaints to avoid penetrations through differentiable optimization, which fails to ensure physical stability, as stated in their paper (Sec. 5.1, page 8 in CAST): `Note that our approach (CAST) does not model full dynamics. For example, an object may not remain stable in its current pose over time.`
> > >
> > > In contrast, HoloScene adopts the geometry energy (Eq. 4) to avoid penetrations, and further leverages physical simulator as critic (Eq. 5) to score and select the best sample to **achieve long-term physical stability in the simulator**.
> > >
> > > ### **II. Rendering and geometry consistency with input: CAST &#x2718;; HoloScene &#x2713;**
> > >
> > > CAST uses conditional generation to process image input and does **not** explicitly enforce appearance or geometry consistency with the input. It relies entirely on direct feed-forward 3D generation capacity to preserve content. As a result, despite semantically aligned well, fine-grained **misalignments are clearly visible in Fig. 2 of CAST** (e.g., the beach rug’s texture and the straw hat’s geometry).
> > >
> > > In contrast, HoloScene **explicitly optimizes** both appearance and geometry consistency through accurate re-rendering (see Eq. 2). This difference in formulation and inference fundamentally distinguishes our approach from CAST and from the broader “Digital Cousin” line of research, where observed images cannot be reliably reproduced. Therefore, HoloScene goes well beyond `a multi-view version of CAST with some engineering integrations` as CAST does not even pursue the goal of re-rendering.
> > >
> > > ### **III: Inference: CAST &rarr; multi-step pipeline; HoloScene &rarr; takes all energy terms as a whole**
> > >
> > > CAST follows a pipelined framework that combines iterative feed-forward object generation and geometry alignment (Sec. 4), and physics-aware correction to avoid collisions (Sec. 5). It performs **greedy optimization**, solving each subgoal sequentially. E.g. after the iterative object generation and alignment step, object geometry is fixed, and only object pose is updated in physics-aware correction, hence supporting physics won't update shape.
> > >
> > > In contrast, HoloScene takes the energy formulation in Eq. 2 as a whole and **jointly optimize** via inverse rendering as well as generative sampling and scoring according to the overall entire energy for every sample explicitly, so that all our goals above are wholly optimized.
> > >
> > > Therefore, we kindly disagree with the reviewer to judge HoloScene as a purely `system-level integration`, `minor strategic variation rather than a new paradigm`, and `highly similar to those of CAST`.
> > >
> > > **NOTE:** The `generative alignment for handling multi-modality` is NOT `analogous` to the **generative sampling** in HoloScene: the `generative alignment` is for `pixel-aligned geometry` in feed forward object generation as stated in CAST's paper, but **generative sampling** is for optimization towards the minimization of all energy terms in HoloScene.

---

> > > > ### Author Response · Authors · 2025-08-05
> > > > **(2/2) Response to Reviewer Np4v's Follow-Up Comments**
> > > >
> > > > ### **(Continue)**
> > > >
> > > > ---
> > > >
> > > > # **Comparisons with DRAWER:**
> > > >
> > > > ## **DRAWER has a different goal from HoloScene.**
> > > >
> > > > As the reviewer correctly points out in the response, `DRAWER’s primary evaluation axes` lie in `articulation accuracy and motion plausibility`.
> > > >
> > > > However, HoloScene focuses on **rigid-body reconstruction with physical plausibility**. Per reviewer request, we have included comparison with DRAWER on rigid-body geometry, rendering realism, and physical stability, which are within our scope, to distinguish the performance in the regard of rigid object in DRAWER.
> > > >
> > > > Since HoloScene does not model articulation, evaluating it using DRAWER’s primary metrics (e.g., articulation accuracy) would be misaligned with our stated objectives. That said, for completeness, we are happy to report DRAWER’s articulation metrics for HoloScene and include them in our revised paper with `0%` articulation accuracy and `+inf` articulation motion error, as expected for a method not designed for articulated modeling. We can include these numbers for transparency and completeness if helpful, though we believe this does not diminish our contributions to physically plausible rigid-body reasoning.
> > > >
> > > > Although `DRAWER already includes geometry completion for rigid objects as part of its pipeline`, we do not view this as diminishing our contribution—**many prior works [46, 47, 71, 73] also include geometry completion**. As we confirmed with the authors, DRAWER’s use of generative completion is limited to **completing the interior of drawers and cabinets**, in line with its primary focus on articulation. Its rigid-body completion relies on **simple SDF isosurface extraction** (see Supp. Sec. C.1 of DRAWER). In contrast, HoloScene leverages **generative shape modeling** as well as **generative inpainting** to more effectively complete **invisible rigid-body geometry**, achieving superior performance compared to DRAWER, as shown in the evaluation table included in our previous rebuttal. Moreover, as we have previously emphasized, several key components of HoloScene—including **penetration-free modeling**, **physical stability**, and **hierarchical scene graph reasoning**—are **not supported in DRAWER**. Therefore, we believe HoloScene does `constitute a substantial methodological difference` with respect to DRAWER.
> > > >
> > > >
> > > > # **Comparison Table:**
> > > >
> > > >
> > > > To summarize, we include a comparison table below for reviewers' and AC's reference.
> > > >
> > > > | Method    | Input modality    | Geometry completeness | Penetration free | Rendering & Geometry consistency with input | Physical stability | Articulation |
> > > > | --------- | --- | --------------------- | ---------------- | ------------------------------------------- | ------------------ | ------------ |
> > > > | CAST      |  Image   | &#x2713;              | &#x2713;         | &#x2718;                                    | &#x2718;           | &#x2718;     |
> > > > | HoloScene | Video    | &#x2713;              | &#x2713;         | &#x2713;                                    | &#x2713;           | &#x2718;     |
> > > > | DRAWER    | Video    | &#x2713;              | &#x2718;         | &#x2713;                                    | &#x2718;           | &#x2713;     |

---

### Official Review · Reviewer_x1eu · 2025-06-25

**Clarity:** 2
**Significance:** 3
**Originality:** 3
**Rating:** 4
**Confidence:** 4

**Summary:**

The study (HoloScene) presents a novel framework for reconstructing interactive, physically plausible 3D scenes from a single RGB video. HoloScene represents a scene as a structured 3D scene graph, where each node encodes object geometry (via neural SDFs and meshes), appearance (using Gaussian splats), and physical properties (mass, friction, etc.). Edges in the graph capture inter-object relationships (e.g., support or contact), enabling hierarchical and physically grounded interactions. The reconstruction is formulated as an energy-based optimization problem balancing observation fidelity (RGB, depth, masks), shape completeness, geometric consistency, and physical stability. The authors propose a three-stage pipeline: Initializing via gradient-based optimization to match observed inputs; Then diverse object completions are generated using a diffusion model (Wonder3D) and combined via a physics-aware search for plausible layouts; Finally enhance visual realism and ensure consistency.

**Questions:**

Please refer to the weakness comments.

**Ethical Concerns:**

["NO or VERY MINOR ethics concerns only"]

**Final Justification:**

Thank you to the authors for the detailed rebuttal and comprehensive failure case analysis, which addresses many of my concerns regarding the model’s reliability and robustness. The quantitative comparisons on geometry and physics failures are particularly helpful.

My remaining concerns primarily relate to:

1. Computational efficiency – The reconstruction time of ~8 hours per scene may pose challenges for scaling to large datasets, especially when compared with generative methods such as Holodeck, which can produce physically plausible scenes within minutes. While I acknowledge the fidelity advantages of HoloScene, a more explicit discussion of this trade-off and potential paths for improving efficiency would strengthen the paper.

2. Comprehensive baseline comparison – While the current comparisons cover several baselines, I recommend also including results against recent state-of-the-art reconstruction methods—both multi-stage approaches (e.g., CAST, DRAWER) and end-to-end methods (e.g., MIDI-3D)—evaluated on geometry and visual quality. This would more fully position HoloScene within the broader landscape of reconstruction approaches.

I would like to increase my rating to 'borderline accept' considering all above-mentioned aspects.

**Limitations:**

yes

**Paper Formatting Concerns:**

The format is good.

**Quality:**

2

**Strengths And Weaknesses:**

**Strengths**:

***Unified Framework***: HoloScene integrates geometry reconstruction, appearance modeling, and physical simulation into a single coherent pipeline, enabling interactive and simulation-ready 3D scenes from just a video.

***Energy-based Optimization***: The combination of diffusion-based shape completion with physics-guided tree search is novel to ensures visual quality and physical plausibility.

**Weaknesses**:

***Limited Literature Review***: The paper lacks a thorough discussion of related works in interactive 3D scene generation. Several recent generation-based methods—such as Holodeck, Digital Cousins, and Scenethesis—as well as reconstruction-based approaches like DeepPriorAssembly and Architect, have addressed similar goals. Many of these works consider physical plausibility, such as collision avoidance or stability constraints (e.g., Scenethesis considers both). A more comprehensive comparison would help position HoloScene more clearly within the landscape of prior research and highlight its unique contributions.

***Insufficient Baseline Comparisons***: The experimental evaluation relies on relatively outdated baselines, such as ObjSDF++ (2023), which has already been compared with PhyRecon and DP-Recon. More recent state-of-the-art methods, especially those mentioned above, should be included in the comparisons or at least discussed to strengthen the empirical claims.

***Efficiency and Reliability Concerns***: The proposed pipeline involves multiple modular components, including segmentation models, inpainting methods, and generative 3D models. However, the paper does not analyze the reliability of this multi-stage approach or present failure case studies. Furthermore, computational cost and runtime efficiency are not reported, which raises questions about the practical feasibility of the method in real-world or time-sensitive applications.

***Methodological Clarity***:

1. The regularization terms in the loss function (e.g., completeness, geometry, and physics terms) are under-explained. A more detailed formulation and intuition behind each term would improve clarity.

2. The criteria for distinguishing background scene vs. foreground objects are not clearly defined. For example, are wall-mounted items considered part of the background?

3. The Object Reconstruction (OR) metric is vaguely described. It remains unclear how the object/scene completeness is quantitatively evaluated—particularly in cases of occlusion or partial visibility.

Reference:
[1] Holodeck: Language Guided Generation of 3D Embodied AI Environments
[2] Digital Cousins: Automated Creation of Digital Cousins for Robust Policy Learning
[3] Scenethesis: A Language and Vision Agentic Framework for 3D Scene Generation
[4] Architect: Generating Vivid and Interactive 3D Scenes with Hierarchical 2D Inpainting
[5] DeepPriorAssembly: Zero-Shot Scene Reconstruction from Single Images with Deep Prior Assembly

---

> ### Author Rebuttal · Authors · 2025-07-30
>
> We thank the reviewer for recognizing our unified framework and energy-based optimization design for simulation-ready environment reconstruction.
>
> We now address the concerns.
>
> **Literature review:**
> We discuss all the papers mentioned in the review, and will include them in the final version.
> * **Digital Cousins [1]:**
> In the first row of Table 1, we include Digital Cousins (ACDC) [1], a single-image reconstruction-by-retrieval approach that operates under a fundamentally different setting than HoloScene. It retrieves plausible 3D assets based on similar semantics and layouts rather than precisely reconstructing the actual scene geometry and appearance. As such, it is not directly comparable to our method under the rendering or reconstruction metrics we report, since it does not aim to faithfully replicate the original scene. In contrast, HoloScene is designed to balance both simulation capability (as explored in Digital Cousins) and scene fidelity (as targeted by reconstruction methods).
> * **DeepPriorAssembly [2]:**
> DeepPriorAssembly [2], like Digital Cousins [1], is a single-image reconstruction method and is not designed to faithfully reconstruct scenes consistent with image sets or video input—the primary focus of HoloScene. In Table 1 and the related work section, we already discuss methods with similar settings, including Gen3DSR and CAST. We will include DeepPriorAssembly [2] in the final version for completeness.
> * **Scenethesis [4]:**
> Scenethesis is a text-conditioned scene **generation** method, with a goal fundamentally different from ours. HoloScene focuses on reconstructing simulatable digital twins from images or videos. We also note that Scenethesis appeared on arXiv on May 5, 2025. According to the NeurIPS 2025 FAQ, *“Papers appearing online after March 1st, 2025 are generally considered concurrent to NeurIPS submissions. Authors are not expected to compare to those.”*
> * **Generation / retrieval-based scene generation:**
> Methods such as Holodeck [3], Digital Cousins [1], Scenethesis [4], and Architect [5] generate 3D scenes from text or sparse inputs. While their outputs are also object-centric scene assets, their primary goal is to create diverse and plausible layouts—not to reconstruct high-fidelity digital twins faithful to input images in appearance or geometry. Consequently, these methods are not directly comparable to HoloScene under reconstruction or rendering metrics. In contrast, HoloScene integrates generative priors into a reconstruction pipeline that accurately replicates input videos and produces simulation-ready environments. It further leverages IsaacSim to ensure physical stability (see Eq. 5), as demonstrated in Fig. 4.
>
> We will make these distinctions and contributions more explicit in the final version.
>
> **Baseline comparisons:**
> Thank you for the suggestions. We respectfully disagree with this concern. The methods mentioned in the reviews—Digital Cousins [1], Holodeck [3], Scenethesis [4], and Architect [5]—are retrieval or generation-based and not designed to produce 3D scenes that **align accurately** with input images or videos at *pixel-level appearance* and *centimeter-level geometry*, as elaborated in the previous response. These approaches target a different goal: generating diverse and plausible virtual scenes. While they reflect the state of the art in that domain, they are not representative of the current state of high-fidelity, simulation-ready reconstruction and twinning. As such, they are not directly comparable using our reconstruction or rendering metrics.
>
> In contrast, our selected baselines are task-aligned and up-to-date. DP-Recon [6], included in our experiments, is a recent state-of-the-art method published at CVPR 2025 (June). ObjSDF++ [7], while introduced earlier, shares the same multi-view reconstruction setting as HoloScene and remains a strong and relevant baseline. Along with PhyRecon, these methods reflect the state of the field at the time of submission. We do not believe ObjSDF++ loses its relevance simply because it has been previously compared with DP-Recon.
>
> Overall, we believe our evaluation setup and baseline choice are fair, representative, and meaningful for assessing progress on this task.
>
> **Failure cases analysis:**
> Our paper has shown failure cases caused by imperfections in foundation models, including inpainting from LaMa[8] and completion from Wonder3D[9], which can be found in Fig. 2, 3 in the main paper and Fig. 8 in the supplementary material and videos. We can witness that although Holoscene has achieved superior results over baselines, it still suffers from imperfect foundation model results. We will include a section on failure case analysis. Here is a brief analysis of failure cases:
> * **Inpainting:**
> The inpainting results with LaMa are not necessarily consistent with the background. (See the background inpainting color in the bottom of Fig. 2, and the sofa inpainting color in our object appearance results of Fig. 3)
> * **Completion:**
> The generation results of Wonder3D are not necessarily perfect as well. From Fig. 3, we can see the geometry of the back of the sofa has some artifacts (somewhat concave but not flat), which is due to the imperfect predictions from Wonder3D.
> * **Physics:**
> From Fig. 8 in the supplementary material and the simulation videos in the supplementary material, in the experiments with the iGibson[10] dataset, we observe that some objects still deviate from original poses after applying gravity, which is due to imperfect predictions by LaMa and Wonder3D.
> * **Conclusion:**
> The failure cases are commonly from the foundation model's imperfect results. Holoscene could keep improving performance by leveraging more advanced foundation models.
>
> **Reliability against foundation model failures:**
> HoloScene exhibits robustness to individual foundation model failures by jointly optimizing multiple energy terms and selecting the best among sampled solutions. Failures in one module typically introduce inconsistencies, which are reflected as higher energy, making such samples less likely to be selected. This contrasts with traditional multi-stage pipelines, where a single module failure can propagate downstream and compromise the entire output. This robustness is reflected in both our real-world examples—where foundation models are imperfect—and in benchmark metrics.
>
> That said, this sampling-based robustness is not a guarantee against all types of failures. Catastrophic errors or insufficient sampling coverage, as shown in some of our failure cases, can still lead to imperfect final results.
>
> **Computation efficiency:**
> *Simulation and rendering runtime*: Our simulation-ready environments support high-fidelity physical simulation and rendering at over 100 FPS on a single A6000 GPU. *Reconstruction runtime* The HoloScene reconstruction stage takes an average of 8 hours per scene. For comparison, DP-Recon [6], a state-of-the-art baseline, requires over 14 hours on average for similar reconstruction tasks.
>
> **Regularization terms:**
> We discuss completeness energy (L159-L167), geometry energy (L168-L175), and physics energy (L176-L183) in Sec 3.2. Because of the limit of maximum characters, we will include the intuition and a more detailed formulation for each energy term in the final version.
>
> **Criteria for foreground and background objects:**
> We define foreground objects as those that can be affected by **external forces**, such as gravity. For example, walls and wall-mounted structures (e.g., ceiling lights) are considered background objects, as they are fixed and unlikely to move under simple external forces. In contrast, objects merely leaning against walls—such as bookshelves—are considered foreground, as they can respond to physical interactions.
>
> **Object Reconstruction (OR) metric:**
> Thank you for pointing this out — there may have been a misunderstanding. The OR metric quantifies how many objects are present in the final reconstruction, regardless of their completeness. This metric is not designed to evaluate object completeness quantitatively; instead, the object-level CD, F1, and NC metrics in Table 2 serve that purpose.
>
> In our baseline experiments, some small objects are missing from the reconstructions because no valid SDFs exist for these objects, preventing mesh extraction via marching cubes. This occurs due to their limited appearance in training images and the constraints of certain loss terms (such as the SDS loss in DP-RECON). In contrast, HoloScene addresses this issue by implementing balanced sampling across instances, ensuring robust reconstruction of all objects.
>
> Examples of missing objects in baseline reconstructions can be found in Figures 5-8 in the supplementary material. Figure 7 (the "no physics" column) shows DP-Recon reconstructing only one object on the shelf, while Figure 8 demonstrates that all baselines fail to reconstruct the small vase.
>
>
> [1] Dai, Tianyuan, et al. "Automated creation of digital cousins for robust policy learning."
>
> [2] Zhou, Junsheng, et al. "Zero-shot scene reconstruction from single images with deep prior assembly."
>
> [3] Yang, Yue, et al. "Holodeck: Language guided generation of 3d embodied ai environments."
>
> [4] Ling, Lu, et al. "Scenethesis: A language and vision agentic framework for 3d scene generation."
>
> [5] Wang, Yian, et al. "Architect: Generating vivid and interactive 3d scenes with hierarchical 2d inpainting."
>
> [6] Ni, Junfeng, et al. "Decompositional neural scene reconstruction with generative diffusion prior."
>
> [7] Wu, Qianyi, et al. "Objectsdf++: Improved object-compositional neural implicit surfaces."
>
> [8] Suvorov, Roman, et al. "Resolution-robust large mask inpainting with fourier convolutions."
>
> [9] Long, Xiaoxiao, et al. "Wonder3d: Single image to 3d using cross-domain diffusion."
>
> [10] Li, Chengshu, et al. "igibson 2.0: Object-centric simulation for robot learning of everyday household tasks."

---

> ### Comment · Reviewer_x1eu · 2025-08-06
> **Still unclear about the reliability and robustness of the model**
>
> I appreciate the author's effort to differentiate the current generative method-based SOTA from Holoscene. While I remain somewhat unconvinced by the method's reliability and robustness, the authors have demonstrated its advancements through comparisons with several baselines and a discussion of CAST. Since these methods aim to achieve similar goals, it is important to include a statistical analysis of Holoscene’s failure rate and a direct comparison with the current reconstruction-based SOTA. Additionally, considering that Holoscene requires a significant amount of time to reconstruct a scene (averaging 8 hours per scene), it would be helpful to clarify its advantages over mentioned generative 3D scene methods.

---

> > ### Author Response · Authors · 2025-08-07
> > **(1/3) Response to Reviewer x1eu's Follow-Up Comments**
> >
> > We thank the reviewer for the follow-up comments and suggestions.
> >
> > ## **Quantitative Failure Rate Comparison**
> >
> > Following your suggestion, we provide a `statistical analysis of HoloScene’s failure rate` and a `direct comparison with current reconstruction-based SOTA` methods that `aim to achieve similar goals`, including ObjSDF++, PhyRecon, and DP-Recon.
> >
> > We quantify both *geometry failures* and *physics failures*—the most common failure modes in simulation-ready asset reconstruction—and identify typical causes for each. This provides a clear root-cause analysis of where and how our method improves over prior work, highlighting the reliability and robustness of our framework.
> >
> > ### **Geometry failure**
> >
> > **Definition**. For this study, a predicted geometry is considered failed if its F-Score (F1) is below 70, using a 0.05 m threshold compared to the GT geometry.
> >
> > **Reason and Phenomenon**. Geometry failures are mainly caused by:
> > * *Invalid object surfaces* — surface shape is null, non-manifold or non-watertight, often due to noisy observations, poor optimization, or insufficient regularization, leading to invalid signed distance fields.
> > * *Partial shape* — caused by the absence of shape priors (e.g., ObjectSDF++ and PhyRecon) or imperfect priors (e.g., DP-Recon, HoloScene).
> > * *Oversized shape* — typically resulting from imperfect priors or under-optimized physical constraints.
> >
> > **Evaluation**. We evaluate geometry failures on the iGibson sub-dataset. The table below summarizes the the results.
> >
> > |  Method   | Total objects # | Failure # &darr; | Failure rate % &darr; | Invalid geometry # | Partial shape # | Oversized shape # |
> > |:---------:|:---------------:|:----------------:|:---------------------:|:---------------:|:------------------:|:--------------------:|
> > | ObjSDF++  |       97        |        52        |         53.6          |       34        |         13         |          **5**           |
> > | PhyRecon  |       97        |        64        |         66.0          |       36        |         21         |          7          |
> > | DP-Recon  |       97        |        70        |         72.2          |       25        |         33         |          12          |
> > | HoloScene |       97        |      **24**      |       **24.7**        |        **0**        |         **13**         |          11          |
> >
> >
> > The results reveal several key insights:
> > - **Generative priors reduce invalid geometry**:
> >   Methods using generative shape priors (DP-Recon, HoloScene) significantly reduce *invalid geometry* cases. Unlike ObjSDF++ and PhyRecon, which lack such priors, DP-Recon uses a SDS-based loss that helps but still sometimes produces invalid shapes due to optimization challenges. In contrast, HoloScene achieves **zero invalid cases**, demonstrating high reliability through balanced sampling across all instances as well as generative inpainting and sampling.
> > - **Sampling improves shape accuracy**:
> >   Despite using SDS loss, DP-Recon performs worse on *partial* and *oversized* shapes than ObjSDF++ and PhyRecon, likely due to the limitations of SDS in completing shapes. HoloScene leverages sampling-based generative inference combined with physical constraints and energy-based scoring. While it doesn’t eliminate all size errors, it offers a strong overall performance.

---

> > > ### Author Response · Authors · 2025-08-07
> > > **(2/3) Response to Reviewer x1eu's Follow-Up Comments**
> > >
> > > ### **(Continue)**
> > >
> > > ---
> > >
> > > ### **Physics failure**
> > >
> > > **Definition**. The reconstructed object has a movement over 0.05 of the scene unit length or rotates over 10 degrees in the simulator after applying gravity in the composed scene (the same threshold as we evaluate stability in paper).
> > >
> > > **Reason and Phenomenon**. Physics failures are mainly caused by:
> > >
> > > * *Invalid shapes* — Missing or incomplete surfaces make objects incompatible with the physics simulator.
> > > * *Overlooked contact pairs* — Some methods (e.g., PhyRecon) focus only on object-ground contacts (as stated in Supp D.1: `focused solely on object-ground support for simplicity and training efficiency`), ignoring object-object interactions. This weakens physical supervision (PhyRecon Sec. 3.2).
> > > * *Penetration* — Inter-object geometry penetration causes repelling forces during simulation, leading to instability.
> > > * *Drifting* — Inaccurate shape modeling at contact points (even with minimal penetration) can cause gradual drifting from the original position.
> > >
> > >
> > > **Evaluation**. We evaluate physics failures across **all datasets** used in the paper: Replica, ScanNet++, and iGibson. The table below summarizes the `physics failure rate` for each dataset.
> > >
> > > |  Method   | Replica % &darr; | Scannet++ % &darr; | iGibson % &darr; |
> > > |:---------:|:----------------:|:------------------:|:----------------:|
> > > | ObjSDF++  |       60.6       |        71.8        |       53.6       |
> > > | PhyRecon  |       94.4       |        90.6        |       66.0       |
> > > | DP-Recon  |       91.5       |        90.6        |       72.2       |
> > > | HoloScene |     **18.3**     |      **29.4**      |     **24.7**     |
> > >
> > > To better understand the sources of failure, we also break down the failure counts by underlying cause:
> > >
> > > |  Method   | Total object # | Failure # &darr; | Failure rate % &darr; | Invalid shapes # | Overlooked contact pairs # | Penetration # | Drifting # |
> > > |:---------:|:--------------:|:----------------:|:---------------------:|:-------:|:-------------------:|:-----------:|:--------:|
> > > | ObjSDF++  |      253       |       166        |         65.6          |   38    |          0          |     76      |    52    |
> > > | PhyRecon  |      253       |       236        |         93.3          |   58    |         121         |     33      |    24    |
> > > | DP-Recon  |      253       |       235        |         92.9          |   64    |          0          |     102     |    69    |
> > > | HoloScene |      253       |      **66**      |       **26.1**        |    0    |          0          |      6      |    60    |
> > >
> > > The results reveal several key insights into physics failure across methods:
> > > - **Failure Tied to Physics Modeling**:
> > >   ObjSDF++ and DP-Recon often fail due to *invalid shapes* and *penetrations*, stemming from limited or no physics modeling. PhyRecon includes physics but suffers from high failure rates by considering only object-ground contact, neglecting object-object interactions (PhyRecon Supp D.1).
> > > - **Shape Priors vs. Physical Plausibility**:
> > >   DP-Recon emphasizes visual completeness with shape priors but lacks physical stability, leading to drifting and penetration. PhyRecon ensures ground contact but doesn't generalize to more complex interactions.
> > > - **HoloScene’s Integrated Solution**:
> > >   HoloScene combines *generative priors*, *scene graph reasoning*, and a *simulator-as-critic* to model object-object contact and enforce physical plausibility. This reduces penetration failures and eliminates invalid or oversimplified geometry.
> > >
> > > These findings suggest that physical consistency should not be treated as an afterthought—whether through *post-hoc correction* or *surrogate loss terms*—but instead integrated directly into the generative reconstruction process via *hard constraints* and *physical validation*. Joint reasoning over structure, geometry, and physics emerges as a promising direction for building more robust, simulation-ready digital assets. We thank the reviewer again for the nice suggestion.

---

> > > > ### Author Response · Authors · 2025-08-07
> > > > **(3/3) Response to Reviewer x1eu's Follow-Up Comments**
> > > >
> > > > ### **(Continue)**
> > > >
> > > > ---
> > > >
> > > > ## **Discussions over Generative 3D Scene Methods**
> > > >
> > > > Compared to prior `generative 3D scene methods`—Digital Cousins, Holodeck, Scenethesis, and Architect—HoloScene excels in **accurately aligning pixel-level appearance and centimeter-level geometry from input video**, while also ensuring **physical stability**. This sometimes comes at the cost of longer reconstruction time but it enables high-fidelity, simulation-ready digital twins suitable for a wide range of applications.
> > > >
> > > > ### **Digital Cousins**
> > > > Digital Cousins is a retrieval-based method that selects plausible 3D assets based on semantic similarity and layout, rather than reconstructing the actual scene. It prioritizes functional simulation over fidelity. In contrast, HoloScene strikes a balance between **simulation capability** and **faithful scene reconstruction**, including fine-grained geometry and appearance.
> > > >
> > > > ### **Holodeck, Scenethesis, and Architect**
> > > > These methods focus on *text-conditioned* 3D scene generation, aiming to synthesize diverse, plausible indoor scenes. Their goal is fundamentally different: creative generation rather than faithful reconstruction. HoloScene, by contrast, is designed for *reconstructing simulatable digital twins* from real-world input.
> > > >
> > > > In summary:
> > > > - **Generative 3D methods** are best suited for creating imaginative and diverse layouts from text prompts. They are ideal for diversity-driven downstream tasks such as virtual world generation and training embodied agents in varied sim environments.
> > > > - **HoloScene** is better suited for reconstructing **physically plausible, visually consistent** scenes from real-world video. It enables high-fidelity applications requiring a digital twin, including reliable training, validation, and verification for embodied agents in more realistic environments, immersive computing (e.g., XR), and video effects or interactive editing.
> > > >
> > > >
> > > > We will revise related work to reflect the above discussions.

---

### Official Review · Reviewer_nTyK · 2025-07-03

**Clarity:** 3
**Significance:** 4
**Originality:** 3
**Rating:** 5
**Confidence:** 4

**Summary:**

The paper introduces HoloScene, a method for generating interactive, simulation-ready 3D scenes from videos of static environments. The approach combines generative sampling and scene graph-based tree search with energy-based optimization to ensure realistic appearance, geometric completeness, and physical plausibility. The authors evaluate their method on multiple datasets (iGibson, ScanNet++, and Replica), and compare against several object- and scene-level reconstruction baselines (e.g., PhyRecon, ObjSDF++, DP-Recon), demonstrating superior reconstruction quality.

**Questions:**

1.	Could the authors show example input videos (how dense/sparse they are)? What is the typical length? Also, how does the method work if given unstructured image collections instead of a video?
2.	Why is the reconstruction error lower at object level for iGibson, but not at scene level? Some insight would be helpful.
3.	Balancing Training Samples (L268): The idea of balancing samples across all instances—especially to improve recovery of small objects—seems important, but it is not sufficiently explained.
4.	Occlusion Inpainting (L164): How are occluded regions detected for inpainting with LaMa? Is this step performed after reconstructing with Wonder3D?
5.	There appears to be some blurring in the rendered appearance compared to standard 3D Gaussian Splatting (3DGS). What limits the realism here, and how might it be improved?
These could help to improve the exposition (and my score :P)

**Ethical Concerns:**

["NO or VERY MINOR ethics concerns only"]

**Final Justification:**

I had minor comments and questions, which the authors addressed well. I carefully considered the rebuttal and other reviews/rebuttals. Overall, the method is quite comprehensive, and I believe that technical contributions and evaluations are sufficient for publication.

**Limitations:**

Yes

**Paper Formatting Concerns:**

Minor concern - The authors have removed the checklist guidelines from the NeurIPS submission checklist. As far as I know, the original guidelines are expected to remain in the final submission.

**Quality:**

4

**Strengths And Weaknesses:**

The work is quite impressive. For example, it is

1.	Comprehensive: The method considers most aspects of 3D world, addressing both scene-level and object-level reconstruction, producing physically plausible and high-quality outputs while supporting simulation and real-time rendering.

2.	Novelty: The proposed divide-and-conquer strategy, which integrates sampling-based tree-structured search with gradient-based refinement, is elegant and effective.

3.	Strong Results: HoloScene achieves superior performance across geometry accuracy, real-time rendering, and physical plausibility for downstream tasks, with broad evaluations across diverse datasets.

4.	Presentation Quality: The paper is well-written and well-structured, making the core ideas and technical contributions easy to follow.

There are no major weaknesses, but I have a few minor concerns and suggestions for clarification: The paper employs multiple geometry representations (meshes, SDFs, and Gaussians), but the motivation and integration of these components is not clearly explained: Why is an SDF representation needed, given that meshes are required for simulation and Gaussians for rendering? How are SDFs represented—globally per object or truncated SDF? Are the SDF and mesh reconstructions synchronized (e.g., via Marching Cubes)? Overall, the SDF/Mesh/Gaussian is a bit sloppily written, or at least missing motivation for SDF. It’s also unclear if the reconstruction geometry is evaluated with mesh/SDF.

---

> ### Author Rebuttal · Authors · 2025-07-30
>
> We thank the reviewer for appreciating the elegant and comprehensive design and superior performance of the simulation-ready environment.
>
> We now address the concerns.
>
> **Why is an SDF representation needed?**
> Directly optimizing surface meshes from video is challenging due to their complex and discrete topology, especially in large indoor scenes. Neural SDFs provide a flexible, continuous representation that simplifies optimization and has demonstrated superior geometric quality [6, 7]. Once optimized, we extract meshes via marching cubes for simulation and attach Gaussians for high-quality rendering. This design balances reconstruction accuracy, topological flexibility, and downstream usability.
>
> **Are SDFs and meshes synchronized?**
> *Yes*. Meshes are extracted from the SDF using marching cubes (L199–L206), ensuring they are fully synchronized.
>
> **Is geometry evaluated with meshes or SDFs?**
> In Table 2, we evaluate geometry quality using the *final reconstructed mesh*, following standard protocols used by our baselines. This ensures fair comparison across diverse approaches and accurately reflects downstream performance in simulation and other mesh-dependent tasks.
>
> **Could you show example input videos (how dense/sparse they are)?**
> Yes! We will include example input videos in the supplementary video and project website to illustrate capture density. Real-world inputs from ScanNet++ [2] are 1–2 minute sequences at 30 FPS with smooth handheld trajectories. Our synthetic datasets (Replica and iGibson) follow the same setup with comparable frame density and motion patterns.
>
> **What is the typical length?**
> The table below summarizes the average number of frames across different datasets. Our collected videos follow the capture speed and density of ScanNet++ [2]. Typical sequence lengths are several hundred frames for small indoor scenes (Replica, ScanNet++) and over a thousand frames for larger scenes (iGibson). For data we collected ourselves, we used handheld RGB cameras with smooth trajectories at 30 FPS, consistent with ScanNet++ practices. HoloScene is designed to handle input videos of varying lengths with ease. For additional statistics and visualizations, please refer to the ScanNet++ website.
>
> | Dataset | Replica[1] | Scannet++[2] |  iGibson[3]   |
> | -------- | -------- | -------- | --- |
> | Average frame #     | 638     | 365     |   1626  |
>
> **Can HoloScene handle unstructured image collections?**
> Yes! HoloScene supports unstructured image collections as input. The main difference lies in scene graph construction (L193–198): for videos, we sample keyframes and estimate relationships using a VLM, while for unstructured image collections, we can compute pairwise camera distances and incrementally build the scene graph. The remaining inference pipeline remains unchanged (L198).
>
> **Why is the reconstruction error at object level lower than scene level on iGibson?**
> This difference is due to scale and sampling resolution. Our evaluation follows standard geometry protocols [8, 9], which compute metrics by sampling points on surface geometry. At the object level, denser sampling within smaller regions increases the likelihood of finding closer correspondences, leading to lower errors. In contrast, scene-level evaluations span larger and more complex spaces, resulting in higher average error. Therefore, object- and scene-level errors are not directly comparable. That said, within each setting, the relative rankings across baselines still reliably reflect reconstruction quality.
>
> **Explanation of the idea of balancing samples across instances:**
> In Stage 1 (L199–L206), we sample an equal number of pixels for each instance in every optimization iteration. This ensures that smaller objects receive sufficient attention during training and are not overshadowed by larger instances (see L203 and L268).
>
> **How is occlusion inpainting done?**
> We first obtain reconstructed geometry for each object after Stage 1 inference (L199). Then, LaMa takes the rendered instance masks as input to inpaint occluded regions (L164). The inpainted images are subsequently passed to Wonder3D to generate novel views for invisible regions.
>
> **Question about rendering quality and potential improvements:**
> We adopt the Gaussian-on-Mesh (GoM) approach from DRAWER [4]. Unlike standard 3D Gaussian Splatting, GoM constrains each Gaussian’s position, orientation, and scale based on the underlying mesh. While this better supports physical simulation and improves scene editability, it reduces the flexibility of Gaussian optimization compared to unconstrained 3DGS, which often trades geometry accuracy for rendering quality. This trade-off explains the slightly blurrier appearance, as also reflected in the rendering metrics in Table 2. Similar limitations are observed in other GoM-based methods such as DRAWER [4] and GaMeS [5]. Rendering quality can be improved by densifying the Gaussians or relaxing mesh constraints, though this may reduce the alignment between appearance and geometry—potentially harming simulation fidelity.
>
> [1] Straub, Julian, et al. "The replica dataset: A digital replica of indoor spaces." arXiv preprint arXiv:1906.05797 (2019).
>
> [2] Yeshwanth, Chandan, et al. "Scannet++: A high-fidelity dataset of 3d indoor scenes." Proceedings of the IEEE/CVF International Conference on Computer Vision. 2023.
>
> [3] Li, Chengshu, et al. "igibson 2.0: Object-centric simulation for robot learning of everyday household tasks." arXiv preprint arXiv:2108.03272 (2021).
>
> [4] Xia, Hongchi, et al. "Drawer: Digital reconstruction and articulation with environment realism." Proceedings of the Computer Vision and Pattern Recognition Conference. 2025.
>
> [5] Waczyńska, Joanna, et al. "Games: Mesh-based adapting and modification of gaussian splatting." arXiv preprint arXiv:2402.01459 (2024).
>
> [6] Yariv, Lior, et al. "Bakedsdf: Meshing neural sdfs for real-time view synthesis." ACM SIGGRAPH 2023 conference proceedings. 2023.
>
> [7] Li, Zhaoshuo, et al. "Neuralangelo: High-fidelity neural surface reconstruction." Proceedings of the IEEE/CVF Conference on Computer Vision and Pattern Recognition. 2023.
>
> [8] Wu, Qianyi, et al. "Objectsdf++: Improved object-compositional neural implicit surfaces." Proceedings of the IEEE/CVF International Conference on Computer Vision. 2023.
>
> [9] Ni, Junfeng, et al. "Decompositional neural scene reconstruction with generative diffusion prior." Proceedings of the Computer Vision and Pattern Recognition Conference. 2025.

---

### Decision · Program_Chairs · 2025-09-17

**Decision:**

Accept (poster)

**Comment:**

The paper firstly received mixed ratings, two positive and two negative. After rebuttal, one negative raise his/her score to be positive. Finally, only one reviewer remain negative, where the primary concern is the limited differences between the proposed method and existing method - CAST.

The AC took very carefully reading on all comments and the submission. Although the AC agrees with that the paper is intrinsically a multi-view version of CAST, such an extension is not a trival task. In the literature, many works focused on 3D reconstruction from a single image, and also many other works focus on reconstrucion from multi-view images or a monocular video. The different problem settings usually causes different algorithm design. In the rebuttal, the author also provided many differences side by side.

Thus, the AC recommends acceptance.